# Generalization of Hamiltonian algorithms

**Andreas Maurer**
Computational Statistics and Machine Learning
Istituto Italiano di Tecnologia, 16163 Genoa, Italy
`am@andreas-maurer.eu`

## Abstract

The paper proves generalization results for a class of stochastic learning algorithms. The method applies whenever the algorithm generates an absolutely continuous distribution relative to some a-priori measure and the Radon Nikodym derivative has subgaussian concentration. Applications are bounds for the Gibbs algorithm and randomizations of stable deterministic algorithms as well as PAC-Bayesian bounds with data-dependent priors.

## 1 Introduction

A stochastic learning algorithm $Q$ takes as input a sample $\mathbf{X} = (X_1, ..., X_n) \in \mathcal{X}^n$, drawn from a distribution $\mu$ on a space $\mathcal{X}$ of data, and outputs a probability measure $Q_{\mathbf{X}}$ on a loss-class $\mathcal{H}$ of functions $h : \mathcal{X} \mapsto [0, \infty)$. A key problem in the study of these algorithms is to bound the generalization gap

$$\Delta(h, \mathbf{X}) = \mathbb{E}[h(X)] - \frac{1}{n} \sum_{i=1}^{n} h(X_i) \tag{1}$$

between the expected and the empirical loss of a hypothesis $h$ drawn from $Q_{\mathbf{X}}$. Here we want to generate $h$ *only once* and seek guarantees with high probability as $\mathbf{X} \sim \mu^n$ *and* $h \sim Q_{\mathbf{X}}$. Alternatively one might want a bound on the expectation $\mathbb{E}_{h \sim Q_{\mathbf{X}}}[\Delta(h, \mathbf{X})]$ with high probability in $\mathbf{X} \sim \mu^n$, corresponding to the use of a stochastic hypothesis, where a new $h \sim Q_{\mathbf{X}}$ is generated for every test point. We concentrate on the former question, but many of the techniques presented also apply to the latter, often easier problem.

From Markov's inequality it follows that for $\lambda, \delta > 0$ with probability at least $1 - \delta$ as $\mathbf{X} \sim \mu^n$ *and* $h \sim Q_{\mathbf{X}}$

$$\Delta(h, \mathbf{X}) \leq \frac{\ln \mathbb{E}_{\mathbf{X}} \left[ \mathbb{E}_{h \sim Q_{\mathbf{X}}} \left[ e^{\lambda \Delta(h, \mathbf{X})} \right] \right] + \ln(1/\delta)}{\lambda}, \tag{2}$$

which suggests to bound the log-moment generating function $\ln \mathbb{E}_{\mathbf{X}} \left[ \mathbb{E}_{h \sim Q_{\mathbf{X}}} \left[ \exp(\lambda \Delta(h, \mathbf{X})) \right] \right]$. With such a bound at hand one can optimize $\lambda$ to establish generalization of the algorithm $Q : \mathbf{X} \mapsto Q_{\mathbf{X}}$.

Inequality (2) is relevant to stochastic algorithms in general, and in particular to the Gibbs-algorithm, where $dQ_{\mathbf{X}}(h) \propto \exp(-(\beta/n) \sum h(X_i)) d\pi(h)$ for some inverse temperature parameter $\beta$ and some nonnegative a priori measure $\pi$ on $\mathcal{H}$. The Gibbs algorithm has its origins in statistical mechanics (Gibbs [1902]). In the context of machine learning it can be viewed as a randomized version of empirical risk minimization, to which it converges as $\beta \to \infty$, whenever $\pi$ has full support. The distribution, often called Gibbs posterior (Catoni [2007]), is a minimizer of the PAC-Bayesian bounds (McAllester [1999]). It is also the limiting distribution of stochastic gradient Langevin dynamics (Raginsky et al. [2017]) under rather general conditions. Generalization bounds in expectation are given by Raginsky et al. [2017], Kuzborskij et al. [2019], most recently by Aminian et al. [2021]. Bounds in probability are given by Lever et al. [2013], implicitly by Dziugaite and Roy [2018], and

38th Conference on Neural Information Processing Systems (NeurIPS 2024).

in Rivasplata et al. [2020] following the method of Kuzborskij et al. [2019]. There is also a bound by Aminian et al. [2023], improving on the one in (Lever et al. [2013]).

Bounding $\ln \mathbb{E}_{\mathbf{X}}\left[\mathbb{E}_{h \sim Q_{\mathbf{X}}}\left[\exp\left(\lambda\Delta\left(h,\mathbf{X}\right)\right)\right]\right]$ is also the vehicle (and principal technical obstacle) to prove PAC-Bayesian bounds with data-dependent prior $Q_{\mathbf{X}}$, as pointed out by Rivasplata et al. [2020] (Theorem 1). Such bounds with *data-independent* prior $Q_{\mathbf{X}} = Q$, have an over twenty year old tradition in learning theory, starting with the seminal work of McAllester (McAllester [1999]), Langford and Seeger (Langford and Seeger [2001], Seeger [2002]), see also Guedj [2019]. If the prior is data-independent, the two expectations in $\ln \mathbb{E}_{\mathbf{X}}\left[\mathbb{E}_{h \sim Q}\left[\exp\left(\lambda\Delta\left(h,\mathbf{X}\right)\right)\right]\right]$ can be exchanged, which reduces the analysis to classical Chernoff- or Hoeffding-inequalities. But a dominant term in these bounds, the KL-divergence $KL\left(P,Q\right) := \mathbb{E}_{h \sim P}\left[\ln\left(dP/dQ\right)\left(h\right)\right]$, will be large unless $P$ is well aligned to $Q$, so the prior $Q$ should already put more weight on good hypotheses with small loss. This motivates the use of distribution-dependent priors, and as the distribution is unknown, one is led to think about data-dependent priors. Catoni already considers the data-dependent Gibbs distribution as a prior in a derivation departing from the distribution-dependent Gibbs measure $\propto \exp\left(-\beta\mathbb{E}_{X \sim \mu}\left[h\left(X\right)\right]\right)$ (Lemma 6.2 in Catoni [2003]). Dziugaite and Roy [2017] used a Gaussian prior with data-dependent width and minimized the PAC-Bayes bound for a Gaussian posterior on a multi-layer neural network, obtaining a good classifier accompanied by a non-vacuous bound. This significant advance raised interest in PAC-Bayes with data-dependent priors. The same authors introduced a method to control data-dependent priors based on differential privacy (Dziugaite and Roy [2018]). Recently Pérez-Ortiz et al. [2021] used Gaussian and Laplace priors, whose means were trained directly from one part of the sample, the remaining part being used to evaluate the PAC-Bayes bound. These developments further motivate the search for in-sample bounds on the log-moment generating function appearing above in (2).

We make the following contributions:

- An economical and general method to bound $\ln \mathbb{E}_{\mathbf{X}}\left[\mathbb{E}_{h \sim Q_{\mathbf{X}}}\left[\exp\left(\lambda\Delta\left(h,\mathbf{X}\right)\right)\right]\right]$ whenever the logarithm of the density of $Q_{\mathbf{X}}$ concentrates exponentially about its mean. In particular, whenever $Q_{\mathbf{X}}$ has the Hamiltonian form $dQ_{\mathbf{X}}\left(h\right) \propto \exp\left(H\left(h,\mathbf{X}\right)\right)d\pi\left(h\right)$, then it is sufficient that the Hamiltonian $H$ satisfies a bounded difference condition. The method also extends to the case, when $H$ is only sub-Gaussian in its arguments.

- Applications to the Gibbs algorithm yielding competitive generalization guarantees, both for bounded and sub-Gaussian losses. Despite its simplicity and generality the method improves over existing results on this well studied problem, removing unnecessary logarithmic factors and various superfluous terms.

- Generalization guarantees for hypotheses sampled once from stochastic kernels centered at the output of uniformly stable algorithms, considerably strengthening a previous result of Rivasplata et al. [2018].

## 2    Notation and Preliminaries

For $m \in \mathbb{N}$, we write $[m] := \{1, ..., m\}$. Random variables are written in upper case letters, like $X, Y, \mathbf{X}$ etc and primes indicate iid copies, like $X'$, $X''$, $\mathbf{X}'$ etc. Vectors are bold like $\mathbf{x},\mathbf{X}$, etc. Throughout $\mathcal{X}$ is a measurable space of data with a probability measure $\mu$, and $\mathbf{X}$ will always be the iid vector $\mathbf{X} = (X_1, ..., X_n) \sim \mu^n$ and $X \sim \mu$. $\mathcal{H}$ is a measurable space of measurable functions $h : \mathcal{X} \to [0, \infty)$, and there is a nonnegative a priori measure $\pi$ on $\mathcal{H}$. The measure $\pi$ need not be a probability measure, it could be Lebesgue measure on the space $\mathbb{R}^d$ of parametrizations of $\mathcal{H}$. Averages over $\pi$ will be written as integrals. $\mathcal{P}\left(\mathcal{H}\right)$ is the set of probability measures on $\mathcal{H}$. Unless otherwise specified $\mathbb{E}$ denotes the expectation in $X \sim \mu$ or $\mathbf{X} \sim \mu^n$. All functions on $\mathcal{H} \times \mathcal{X}^n$ appearing in this paper are assumed to have exponential moments of all orders, with respect to both arguments.

For $\mathbf{x} \in \mathcal{X}^n$, $k \in \{1, ..., n\}$ and $y, y' \in \mathcal{X}$ we define the substitution operator $S_y^k$ acting on $\mathcal{X}^n$ and the partial difference operator $D_{y,y'}^k$ acting on functions $f : \mathcal{X}^n \to \mathbb{R}$ by

$$S_y^k\mathbf{x} = (x_1, ..., x_{k-1}, y, x_{k+1}, ..., x_n) \text{ and } D_{y,y'}^k f\left(\mathbf{x}\right) = f\left(S_y^k\mathbf{x}\right) - f\left(S_{y'}^k\mathbf{x}\right).$$

$D_{y,y'}^k$ always refers to the second argument for functions on $\mathcal{H} \times \mathcal{X}^n$. The generalization gap $\Delta\left(h,\mathbf{X}\right)$ is defined as in (1). Sometimes we write $L\left(h\right) = \mathbb{E}\left[h\left(X\right)\right]$ and $\hat{L}\left(h,\mathbf{X}\right) =$

$(1/n) \sum_{i=1}^{n} h(X_i)$, so that $\Delta(h, \mathbf{X}) = L(h) - \hat{L}(h, \mathbf{X})$. A table of notation is provided in Appendix C.

## 2.1 Hamiltonian algorithms

A stochastic algorithm $Q : \mathbf{x} \in \mathcal{X}^n \mapsto Q_{\mathbf{x}} \in \mathcal{P}(\mathcal{H})$ will be called absolutely continuous, if $Q_{\mathbf{x}}$ is absolutely continuous with respect to $\pi$ for every $\mathbf{x} \in \mathcal{X}^n$ and vice versa. We will only consider absolutely continuous algorithms in the sequel. A real function $H$ on $\mathcal{H} \times \mathcal{X}^n$ is called a Hamiltonian for $Q$ (a term taken from statistical physics), if for all $h \in \mathcal{H}$ and all $\mathbf{x} \in \mathcal{X}^n$

$$dQ_{\mathbf{x}}(h) = \frac{e^{H(h,\mathbf{x})} d\pi(h)}{Z(\mathbf{x})} \text{ with } Z(\mathbf{x}) = \int_{\mathcal{H}} e^{H(h,\mathbf{x})} d\pi(h).$$

The normalizing function $Z$ is called the partition function. Every absolutely continuous $Q$ has the canonical Hamiltonian $H_Q(h, \mathbf{x}) = \ln((dQ_{\mathbf{x}}/d\pi)(h))$ (logarithm of the Radon Nikodym derivative) with partition function $Z \equiv 1$, but adding any function $\zeta : \mathcal{X}^n \to \mathbb{R}$ to $H_Q$ will give a Hamiltonian for the same algorithm with partition function $Z(\mathbf{x}) = \exp(\zeta(\mathbf{x}))$. In practice $Q$ is often defined by specifying some Hamiltonian $H$, so $H_Q(h, \mathbf{x}) = H(h, \mathbf{x}) - \ln Z(\mathbf{x})$ in general. If $\pi$ is a probability measure, then $\mathbb{E}_{h \sim Q_{\mathbf{x}}}[H_Q(h, \mathbf{x})]$ is the KL-divergence $KL(Q_{\mathbf{x}}, \pi)$.

A Hamiltonian for the Gibbs algorithm at inverse temperature $\beta$ is

$$H(h, \mathbf{x}) = -\beta \hat{L}(h, \mathbf{x}) = -\frac{\beta}{n} \sum_{i=1}^{n} h(x_i),$$

putting larger weights on hypotheses with small empirical loss. This is the simplest case covered by our proposed method. If there is a computational cost associated with each $h$, it may be included to promote hypotheses with faster execution. We could also add a negative multiple of $\sum_{i<j}(h(x_i) - h(x_j))^2$, so as to encourage hypotheses with small empirical variance. Monte Carlo methods, such as the Metropolis-Hastings algorithm, can be used to sample from such distributions. Often these are slow to converge, which underlines the practical importance of using a single hypothesis generated only once.

If $\mathcal{H}$ is parametrized by $\mathbb{R}^d$ one may also first compute a vector $A(\mathbf{x}) \in \mathbb{R}^d$ with some algorithm $A$ and then sample from an absolutely continuous stochastic kernel $\kappa$ centered at $A(\mathbf{x})$, so a Hamiltonian is $\ln \kappa(h - A(\mathbf{x}))$. In one concrete version the kernel is an isotropic gaussian, and $H(h, \mathbf{x}) = -\|h - A(\mathbf{x})\|^2 / (2\sigma^2)$. Generalization of these methods is discussed in Section 4.2.

If $Q^{(1)}$ and $Q^{(2)}$ are absolutely continuous stochastic algorithms with Hamiltonians $H_1$ and $H_2$ respectively, then an elementary calculation shows that $H_1 + H_2$ is a Hamiltonian for the algorithm $Q$ obtained by sampling from $Q^{(1)}$ with the measure $\pi$ replaced by $Q^{(2)}$. In this way Hamiltonian algorithms of different type can be combined.

## 3 Main results

Let $Q$ be an absolutely continuous stochastic algorithm, $F : \mathcal{H} \times \mathcal{X}^n \to \mathbb{R}$ some function and define

$$\psi_F(h) := \ln \mathbb{E}\left[e^{F(h,\mathbf{X}) + H_Q(h,\mathbf{X}) - \mathbb{E}[H_Q(h,\mathbf{X}')]}\right].$$

Our method is based on the following proposition.

**Proposition 3.1.** *With $Q$, $F$ and $\psi$ as above*

*(i)* $\ln \mathbb{E}_{\mathbf{X} \sim \mu^n} \mathbb{E}_{h \sim Q_{\mathbf{x}}}\left[e^{F(h,\mathbf{X})}\right] \leq \sup_{h \in \mathcal{H}} \psi_F(h).$

*(ii) Let $\delta > 0$. Then with probability at least $1 - \delta$ in $\mathbf{X} \sim \mu^n$ and $h \sim Q_{\mathbf{X}}$ we have*

$$F(h, \mathbf{X}) \leq \sup_{h \in \mathcal{H}} \psi_F(h) + \ln(1/\delta).$$

*(iii) Let $\delta > 0$. Then with probability at least $1 - \delta$ in $\mathbf{X} \sim \mu^n$ we have*

$$\mathbb{E}_{h \sim Q_{\mathbf{x}}}[F(h, \mathbf{X})] \leq \sup_{h \in \mathcal{H}} \psi_F(h) + \ln(1/\delta).$$

*Proof.* (i) With Jensen's inequality

$$\ln \mathbb{E}_{\mathbf{X}\sim\mu^n} \mathbb{E}_{h\sim Q_{\mathbf{x}}} \left[ e^{F(h,\mathbf{X})} \right] = \ln \mathbb{E}_{\mathbf{X}\sim\mu^n} \left[ \int_{\mathcal{H}} e^{F(h,\mathbf{X})+H_Q(h,\mathbf{X})} d\pi(h) \right]$$

$$= \ln \int_{\mathcal{H}} \mathbb{E}_{\mathbf{X}\sim\mu^n} \left[ e^{F(h,\mathbf{X})+H_Q(h,\mathbf{X})-\mathbb{E}[H_Q(h,\mathbf{X}')]} \right] e^{\mathbb{E}[H_Q(h,\mathbf{X}')]} d\pi(h)$$

$$= \ln \int_{\mathcal{H}} e^{\psi_F(h)} e^{\mathbb{E}[H_Q(h,\mathbf{X}')]} d\pi(h)$$

$$\leq \ln \int_{\mathcal{H}} \mathbb{E}\left[ e^{\psi_F(h)} e^{H_Q(h,\mathbf{X}')} \right] d\pi(h) = \ln \mathbb{E}\left[ \int_{\mathcal{H}} e^{\psi_F(h)} e^{H_Q(h,\mathbf{X}')} d\pi(h) \right]$$

$$= \ln \mathbb{E}_{\mathbf{X}\sim\mu^n} \mathbb{E}_{h\sim Q_{\mathbf{x}}} \left[ e^{\psi_F(h)} \right] \leq \sup_{h\in\mathcal{H}} \psi_F(h).$$

(ii) then follows from Markov's inequality (Section A.1). (iii) follows also by Markov's inequality, since $\ln \mathbb{E}_{\mathbf{X}\sim\mu^n} \left[ e^{\mathbb{E}_{h\sim Q_{\mathbf{x}}}[F(h,\mathbf{X})]} \right] \leq \ln \mathbb{E}_{\mathbf{X}\sim\mu^n} \mathbb{E}_{h\sim Q_{\mathbf{x}}} \left[ e^{F(h,\mathbf{X})} \right]$, by Jensen's inequality. $\square$

To see the point of this proposition let $F(h,\mathbf{X}) = \lambda\Delta(h,\mathbf{X})$. Since $\Delta(h,\mathbf{X})$ is centered, $\psi_{\lambda\Delta}$ is of the form $\ln \mathbb{E}_{\mathbf{X}\sim\mu^n} \left[ e^{f(\mathbf{X})-\mathbb{E}[f(\mathbf{X}')]} \right]$. Many concentration inequalities in the literature (McDiarmid [1998], Boucheron et al. [2013]) follow the classical ideas of Bernstein and Chernoff and are derived from bounds on such moment generating functions. If $F(h,\mathbf{X})$ is not centered, we can use Hölder's or the Cauchy-Schwarz inequality to separate the contributions which $F$ and $H_Q$ make to $\psi_F$. The $F$-contribution can be treated separately and the contribution of $H_Q$ can again be treated with the methods of concentration inequalities.

The last conclusion of the proposition is to show that we can always get bounds in expectation from bounds on the exponential moment. In the sequel we only state the stronger un-expected or "disintegrated" results.

Typically the exponential moment bounds for functions of independent variables depend on the function's stability with respect to changes in its arguments. In Section 4.2 a more advanced concentration inequality will be used, but all other results below depend only on the following classical exponential moment bounds. Most of them can be found in McDiarmid [1998], but since the results there are formulated as deviation inequalities, a proof is given in the appendix, Section A.2.

**Proposition 3.2.** *Let $X, X_1, ..., X_n$ be iid random variables with values in $\mathcal{X}$, $\mathbf{X} = (X_1, ..., X_n)$ and $f : \mathcal{X}^n \to \mathbb{R}$ measurable.*

*(i) If $f$ is such that for all $k \in [n]$, $\mathbf{x} \in \mathcal{X}^n$ we have $\mathbb{E}_X \left[ e^{f(S_X^k \mathbf{x})-\mathbb{E}_{X'}[f(S_{X'}^k \mathbf{x})]} \right] \leq e^{r^2}$, then $\mathbb{E}\left[ e^{f(\mathbf{X})-\mathbb{E}[f(\mathbf{X}')]} \right] \leq e^{nr^2}$.*

*(ii) If $D_{y,y'}^k f(\mathbf{x}) \leq c$ for all $k \in [n]$, $y, y' \in \mathcal{X}$ and $\mathbf{x} \in \mathcal{X}^n$, then $\mathbb{E}\left[ e^{f(\mathbf{X})-\mathbb{E}[f(\mathbf{X}')]} \right] \leq e^{nc^2/8}$.*

*(iii) If there is $b \in (0,2)$, such that for all $k \in [n]$ and $\mathbf{x} \in \mathcal{X}^n$ we have $f(\mathbf{x}) - \mathbb{E}_{X'\sim\mu} \left[ f(S_{X'}^k \mathbf{x}) \right] \leq b$, then with $v_k = \sup_{x\in\mathcal{X}^n} \mathbb{E}_{X\sim\mu} \left[ \left( f(S_X^k \mathbf{x}) - \mathbb{E}_{X'\sim\mu} \left[ f(S_{X'}^k \mathbf{x}) \right] \right)^2 \right]$*

$$\mathbb{E}_{\mathbf{X}} \left[ e^{f(\mathbf{X})-\mathbb{E}[f(\mathbf{X}')]} \right] \leq \exp\left( \frac{1}{2-b} \sum_{k=1}^n v_k \right).$$

Notice that the second conclusion is instrumental in the usual proof of McDiarmid's (or bounded-difference-) inequality (McDiarmid [1998]).

## 3.1 Bounded differences

In the simplest case the Hamiltonian satisfies a bounded difference condition as in (ii) above. Then the only minor complication is to show that the logarithm of the partition function inherits this property. This is dealt with in the following lemma.

**Lemma 3.3.** *Suppose $H : \mathcal{H} \times \mathcal{X}^n \to \mathbb{R}$ and that for all $h \in \mathcal{H}$, $k \in [n]$, $y$, $y' \in \mathcal{X}$ and $\mathbf{x} \in \mathcal{X}^n$ we have $D_{y,y'}^k H(h, \mathbf{x}) \leq c$. Let*

$$H_Q(h, \mathbf{x}) = H(h, \mathbf{x}) - \ln Z(\mathbf{x}) \text{ with } Z(\mathbf{x}) = \int_{\mathcal{H}} e^{H(h,\mathbf{x})} d\pi(h).$$

*Then $\forall h \in \mathcal{H}, k \in [n], y, y' \in \mathcal{X}, \mathbf{x} \in \mathcal{X}^n$ we have $D_{y,y'}^k H_Q(h, \mathbf{x}) \leq 2c$.*

*Proof.* This follows from the linearity of the partial difference operator $D_{y,y'}^k$ and

$$
\begin{aligned}
D_{y'y}^k \ln Z(\mathbf{x}) &= \ln \frac{Z\left(S_{y'}^k \mathbf{x}\right)}{Z\left(S_y^k \mathbf{x}\right)} = \ln \frac{\int_{\mathcal{H}} \exp\left(D_{y',y}^k H(h, \mathbf{x})\right) \exp\left(H\left(h, S_y^k \mathbf{x}\right)\right) d\pi(h)}{\int_{\mathcal{H}} \exp\left(H\left(h, S_y^k \mathbf{x}\right)\right) d\pi(h)} \\
&\leq \ln \sup_{h \in \mathcal{H}} \exp\left(D_{y',y}^k H(h, \mathbf{x})\right) \leq c.
\end{aligned}
$$

$\square$

**Theorem 3.4.** *Suppose $H$ is a Hamiltonian for $Q$ and that for all $k \in [n]$, $h \in \mathcal{H}$, $y$, $y' \in \mathcal{X}$ and $\mathbf{x} \in \mathcal{X}^n$ we have $D_{y,y'}^k H(h, \mathbf{x}) \leq c$ and $h(y) \in [0, b]$. Then*

*(i) For any $\lambda > 0$*

$$\ln \mathbb{E}_{\mathbf{X} \sim \mu^n} \mathbb{E}_{h \sim Q_{\mathbf{x}}}\left[e^{\lambda \Delta}\right] \leq \sup_{h \in \mathcal{H}} \psi_{\lambda \Delta}(h) \leq \frac{n}{8}\left(\frac{\lambda b}{n} + 2c\right)^2.$$

*(ii) If $\delta > 0$ then with probability at least $1 - \delta$ as $\mathbf{X} \sim \mu^n$ and $h \sim Q_{\mathbf{X}}$ we have*

$$\Delta(h, \mathbf{X}) \leq b\left(c + \sqrt{\frac{\ln(1/\delta)}{2n}}\right).$$

*Proof.* Using the previous lemma $D_{y,y'}^k\left(\lambda \Delta(h, \mathbf{x}) + H_Q(h, \mathbf{x})\right) \leq (\lambda b/n) + 2c$. Since $\mathbb{E}[\lambda \Delta(h, \mathbf{X})] = 0$, Proposition 3.2 (ii) gives, for any $h \in \mathcal{H}$,

$$\psi_{\lambda \Delta}(h) = \ln \mathbb{E}\left[e^{\lambda \Delta(h,\mathbf{X}) + H_Q(h,\mathbf{X}) - \mathbb{E}\left[H_Q\left(h,\mathbf{X}'\right)\right]}\right] \leq \frac{n}{8}\left(\frac{\lambda b}{n} + 2c\right)^2,$$

and the first conclusion follows from Proposition 3.1 (i) with $F(h, \mathbf{x}) = \lambda \Delta(h, \mathbf{X})$.

From Proposition 3.1 (ii) we get with probability at least $1 - \delta$ as $\mathbf{X} \sim \mu^n, h \sim Q_{\mathbf{X}}$ that

$$\Delta(h, \mathbf{X}) \leq \lambda^{-1}\left(\frac{n}{8}\left(\frac{\lambda b}{n} + 2c\right)^2 + \ln \frac{1}{\delta}\right) = \frac{\lambda b^2}{8n} + \frac{bc}{2} + \frac{nc^2/2 + \ln(1/\delta)}{\lambda}.$$

Substitution of the optimal value $\lambda = \sqrt{(8n/b^2)(nc^2/2 + \ln(1/\delta))}$ and subadditivity of $t \mapsto \sqrt{t}$ give the second conclusion. $\square$

In applications, such as the Gibbs algorithm, we typically have $c = O(1/n)$. The previous result is simple and already gives competitive bounds for several methods, but it does not take into account the properties of the hypothesis chosen from $Q_{\mathbf{X}}$. To make the method more flexible we use the Cauchy-Schwarz inequality to write

$$
\begin{aligned}
\psi_F(h) &= \ln \mathbb{E}\left[e^{F(h,\mathbf{X}) + H_Q(h,\mathbf{X}) - \mathbb{E}\left[H_Q\left(h,\mathbf{X}'\right)\right]}\right] \\
&\leq \frac{1}{2} \ln \mathbb{E}\left[e^{2F(h,\mathbf{X})}\right] + \frac{1}{2} \ln \mathbb{E}\left[e^{2\left(H_Q(h,\mathbf{X}) - \mathbb{E}\left[H_Q\left(h,\mathbf{X}'\right)\right]\right)}\right] \quad (3)
\end{aligned}
$$

and treat the two terms separately. The disadvantage here is an increase of constants in the bounds. The big advantage is that different types of bounds can be combined.

The next result is based on this idea. It is similar to Bernstein's inequality for sums of independent variables.

**Theorem 3.5.** *Under the conditions of Theorem 3.4 define for each $h \in \mathcal{H}$ its variance $v(h) :=$
$\mathbb{E}\left[(h(X) - \mathbb{E}[h(X')])^2\right]$. Then for $\delta > 0$ with probability at least $1 - \delta$ in $\mathbf{X} \sim \mu^n$ and $h \sim Q_{\mathbf{X}}$*

$$\Delta(h, \mathbf{X}) \leq 2\sqrt{v(h)\left(c^2 + \frac{\ln(1/\delta)}{n}\right)} + b\left(c^2 + \frac{\ln(1/\delta)}{n}\right).$$

Similar to Bernstein's inequality the above gives a better bound if the chosen hypothesis has a small variance. For the proof we use the following lemma, which is a direct consequence of Proposition 3.2 (iii).

**Lemma 3.6.** *Assume that for all $h \in \mathcal{H}$ and $x \in \mathcal{X}$, we have $h(x) \in [0, b]$ and $v(h)$ as in Theorem 3.5 above. Let $\lambda$ be a function $\lambda : \mathcal{H} \to (0, n/b)$ and define $F_\lambda : \mathcal{H} \times \mathcal{X}^n \to \mathbb{R}$ by*

$$F_\lambda(h, \mathbf{X}) = \lambda(h)\Delta(h, \mathbf{X}) - \frac{\lambda(h)^2}{1 - b\lambda(h)/n}\frac{v(h)}{n}.$$

*Then for all $h \in \mathcal{H}$ we have $\mathbb{E}\left[e^{2F_\lambda(h, \mathbf{X})}\right] \leq 1$.*

*Proof.* For every $h \in \mathcal{H}$ we have $2b\lambda(h)/n \in (0, 2)$. Also $\forall \mathbf{x} \in \mathcal{X}^n$, and $\forall h \in \mathcal{H}$

$$\Delta(h, \mathbf{x}) - \mathbb{E}_{X' \sim \mu}\left[S_{X'}^k \Delta(h, \mathbf{x})\right] = \frac{1}{n}(\mathbb{E}[h(X')] - h(x_k)) \leq \frac{b}{n}.$$

Thus for every $h \in \mathcal{H}$ we can apply Proposition 3.2 (iii) to $f(\mathbf{x}) = 2\lambda(h)\Delta(h, \mathbf{x})$ and obtain

$$
\begin{aligned}
\mathbb{E}\left[e^{2F(h, \mathbf{X})}\right]^{1/2} &= \mathbb{E}\left[\exp(2\lambda(h)\Delta(h, \mathbf{X}))\right]^{1/2}\exp\left(\frac{-\lambda(h)^2}{1 - b\lambda(h)/n}\frac{v(h)}{n}\right) \\
&\leq \exp\left(\frac{2\lambda(h)^2}{2 - 2b\lambda(h)/n}\frac{v(h)}{n}\right)\exp\left(\frac{-\lambda(h)^2}{1 - b\lambda(h)/n}\frac{v(h)}{n}\right) = 1.
\end{aligned}
$$

$\square$

*Proof of Theorem 3.5.* Let

$$\lambda(h) = \frac{\sqrt{nc^2 + \ln(1/\delta)}}{(b/n)\sqrt{nc^2 + \ln(1/\delta)} + \sqrt{\frac{v(h)}{n}}}$$

and let $F_\lambda$ be as in Lemma 3.6. It follows from Lemma 3.3 and Proposition 3.2 (ii) that for all $h \in \mathcal{H}$ we have $\ln \mathbb{E}\left[e^{2\left(H_Q(h, \mathbf{X}) - \mathbb{E}[H_Q(h, \mathbf{X}')]\right)}\right] \leq 2nc^2$. Then (3) and Lemma 3.6 give $\psi_F(h) \leq nc^2$. Thus from Proposition 3.1 and division by $\lambda(h)$

$$\Pr_{X \sim \mu^n, h \sim Q_{\mathbf{X}}}\left\{\Delta(h, \mathbf{X}) > \frac{\lambda(h)}{1 - b\lambda(h)/n}\frac{v(h)}{n} + \frac{nc^2 + \ln(1/\delta)}{\lambda(h)}\right\} < \delta.$$

Inserting the definition of $\lambda(h)$ and simplifying completes the proof. $\square$

At the expense of larger constants the role of the variance in this result can be replaced by the empirical error, using $v(h) \leq \mathbb{E}[h(X)]^2 \leq b\,\mathbb{E}[h(X)]$ and a simple algebraic inversion, which is given in the appendix, Section A.3.

**Corollary 3.7.** *Under the conditions of Theorem 3.5 for $\delta > 0$ with probability at least $1 - \delta$ in $\mathbf{X} \sim \mu^n$ and $h \sim Q_{\mathbf{X}}$*

$$\Delta(h, \mathbf{X}) \leq 2\sqrt{\hat{L}(h, \mathbf{X})b\left(c^2 + \frac{\ln(1/\delta)}{n}\right)} + 5b\left(c^2 + \frac{\ln(1/\delta)}{n}\right).$$

## 3.2 Subgaussian hypotheses

Some of the above extends to unbounded hypotheses. A real random variable $Y$ is $\sigma$-subgaussian for $\sigma > 0$, if $\mathbb{E}\left[\exp\left(\lambda\left(Y - \mathbb{E}\left[Y'\right]\right)\right)\right] \le e^{\lambda^2\sigma^2/2}$ for every $\lambda \in \mathbb{R}$. The proof of the following result is given in the appendix (Section A.4) and uses ideas very similar to the proofs above.

**Theorem 3.8.** *Let $Q$ have Hamiltonian $H$ and assume that $\forall h \in \mathcal{H}$ there is $\rho(h) > 0$ such that*

$$\forall \lambda \in \mathbb{R},\ \mathbb{E}\left[e^{\lambda\left(h(X) - \mathbb{E}\left[h\left(X'\right)\right]\right)}\right] \le \exp\left(\frac{\lambda^2\rho(h)^2}{2}\right).$$

*Let $\rho = \sup_{h \in \mathcal{H}} \rho(h)$ and suppose that $\forall \lambda \in \mathbb{R}, k \in [n], h \in \mathcal{H}$*

$$\mathbb{E}\left[e^{\lambda\left(H\left(h, S_X^k\mathbf{x}\right) - \mathbb{E}\left[H\left(h, S_{X'}^k\mathbf{x}\right)\right]\right)}\right] \le \exp\left(\frac{\lambda^2\sigma^2}{2}\right).$$

*(i) Then for any $h \in \mathcal{H},\ \lambda > 0$*

$$\ln \mathbb{E}_{\mathbf{X}\sim\mu^n}\mathbb{E}_{h\sim Q_\mathbf{X}}\left[e^{\lambda\Delta}\right] \le \psi_{\lambda\Delta}(h) \le \frac{\left(\lambda\rho(h)/\sqrt{n} + 2\sqrt{n}\sigma\right)^2}{2},$$

*and with probability at least $1 - \delta$ we have as $\mathbf{X} \sim \mu^n$ and $h \sim Q_\mathbf{X}$ that*

$$\Delta(h, \mathbf{X}) \le \rho\left(2\sigma + \sqrt{\frac{2\ln(1/\delta)}{n}}\right).$$

*(ii) With probability at least $1 - \delta$ we have as $\mathbf{X} \sim \mu^n$ and $h \sim Q_\mathbf{X}$ that*

$$\Delta(h, \mathbf{X}) \le \rho(h)\left(\sqrt{32}\sigma + \sqrt{\frac{4\ln(1/\delta)}{n}}\right).$$

The assumptions mean that every hypothesis has its own subgaussian parameter and that the Hamiltonian is subgaussian in every argument if all other arguments are fixed. The first conclusion parallels the bound for Hamiltonians with bounded differences in Theorem 3.4. The second conclusion has larger constants, but scales with the subgaussian parameter of the hypothesis actually chosen from $Q_\mathbf{X}$, which can be considerably smaller, similar to the Bernstein-type inequality Theorem 3.5.

## 4 Applications

### 4.1 The Gibbs algorithm

The Gibbs distribution for a sample $\mathbf{x}$ is $dQ_{\beta,\mathbf{x}}(h) = Z^{-1}\exp\left(-(\beta/n)\sum_{i=1}^n h(x_i)\right)d\pi(h)$, so the Hamiltonian is $H(h, \mathbf{x}) = -(\beta/n)\sum_{i=1}^n h(x_i)$. It is the minimizer of the PAC-Bayesian bounds (McAllester [1999]) as well as the limiting distribution of stochastic gradient Langevin dynamics (Raginsky et al. [2017]), generalization bounds for the Gibbs distribution translate to guarantees for these algorithms. Let us first assume bounded hypotheses, for simplicity $h : \mathcal{X} \to [0, 1]$. Then we can use Theorems 3.4 and 3.5 and Corollary 3.7 with $c = \beta/n$. Theorem 3.4 gives with probability at least $1 - \delta$ in $\mathbf{X} \sim \mu^n$ and $h \sim Q_{\beta,\mathbf{X}}$ that

$$\Delta(h, \mathbf{X}) \le \frac{\beta}{n} + \sqrt{\frac{\ln(1/\delta)}{2n}}. \tag{4}$$

We were not able to find this simple bound in the literature. It improves over

$$\Delta(h, \mathbf{X}) \le \frac{4\beta}{n} + \frac{2 + \ln\left((1 + \sqrt{e})/\delta\right)}{\sqrt{n}}$$

obtained in (Rivasplata et al. [2020], Sec. 2.1 and Lemma 3) not only in constants, but, more importantly, in its dependence on the confidence parameter $\delta$. The principal merit of (4), however, lies in the generality and simplicity of its proof (compare the proof of Lemma 3 in Rivasplata et al. [2020]).

Upon the substitution $c = \beta/n$ Theorem 3.5 leads to a variance dependent bound, for which we know of no comparable result.

From Corollary 3.7 we get for the Gibbs algorithm with probability at least $1 - \delta$ in $\mathbf{X}$ and $h \sim Q_{\beta,\mathbf{x}}$

$$\Delta\left(h, \mathbf{X}\right) \leq 2\sqrt{\hat{L}\left(h, \mathbf{X}\right)\left(\frac{\beta^2}{n^2} + \frac{\ln\left(1/\delta\right)}{n}\right)} + 5\left(\frac{\beta^2}{n^2} + \frac{\ln\left(1/\delta\right)}{n}\right), \tag{5}$$

For hypotheses with small empirical error this approximates a "fast convergence rate" of $O\left(1/n\right)$. Comparable bounds in the literature involve the so-called "little KL-divergence". For two numbers $s, t \in [0, 1]$ the relative entropy of two Bernoulli variables, with means $s$ and $t$ respectively, is $kl\left(s, t\right) = s \ln\left(s/t\right) + \left(1 - s\right) \ln\left(\left(1 - s\right)/\left(1 - t\right)\right)$. Various authors give bounds on $\mathbb{E}_{h \sim Q_{\beta,\mathbf{x}}}\left[kl\left(\hat{L}\left(h, \mathbf{X}\right), L\left(h\right)\right)\right]$ with high probability in the sample. Rivasplata et al. [2020] give

$$\mathbb{E}_{h \sim Q_{\beta,\mathbf{x}}}\left[kl\left(\hat{L}\left(h, \mathbf{X}\right), L\left(h\right)\right)\right] \leq \frac{2\beta^2}{n^2} + \sqrt{2\ln 3}\frac{\beta}{n^{3/2}} + \frac{1}{n}\ln\left(\frac{4\sqrt{n}}{\delta}\right),$$

and there is a similar bound in Dziugaite and Roy [2018] and a slightly weaker one in Lever et al. [2013]. The most useful form of these bounds is obtained using the following inversion rule (Tolstikhin and Seldin [2013], see also Alquier [2021]): if $kl\left(\hat{L}\left(h, \mathbf{x}\right), L\left(h\right)\right) \leq B$ then $\Delta\left(h, \mathbf{x}\right) \leq \sqrt{2\hat{L}\left(h, \mathbf{x}\right)B} + 2B$. If this rule is applied to the $kl$-bound above, it becomes clear, that it is inferior to (5), not only because of the logarithmic dependence on $n$, but also because of artifact terms, which are difficult to interpret, like the superfluous $\beta/n^{3/2}$.

If every $h\left(X\right)$ is $\rho\left(h\right)$-subgaussian and $\rho = \sup_h \rho\left(h\right)$, then by linearity of the subgaussian parameter $H\left(h, \mathbf{X}\right)$ is $\rho\beta/n$-subgaussian in every argument for every $h$, and Theorem 3.8 gives with probability at least $1 - \delta$ in $\mathbf{X} \sim \mu^n$ and $h \sim Q_{\beta,\mathbf{X}}$

$$\Delta\left(h, \mathbf{X}\right) \leq \rho\left(h\right)\left(\frac{4\sqrt{2}\rho\beta}{n} + \sqrt{\frac{4\ln\left(1/\delta\right)}{n}}\right).$$

Recently Aminian et al. [2023] gave a very interesting bound in probability for sub-gaussian hypotheses, which however is not quite comparable to the above, as it bounds the posterior expectation of $\Delta$ and relies on a distribution-dependent prior.

## 4.2 Randomization of stable algorithms

Suppose that $\mathcal{H}$ is parametrized by $\mathbb{R}^d$, with $\pi$ being Lebesgue measure. To simplify notation we identify a hypothesis $h \in \mathcal{H}$ with its parametrizing vector, so that $h$ is simultaneously a vector in $\mathbb{R}^d$ and a function $h : x \in \mathcal{X} \mapsto h\left(x\right) \in \mathbb{R}$. Following Rivasplata et al. [2018] we define the *hypothesis sensitivity coefficient* of a vector valued algorithm $A : \mathcal{X}^n \to \mathbb{R}^d$ as

$$c_A = \max_{k \in [n]} \sup_{\mathbf{x} \in \mathcal{X}^n, y, y' \in \mathcal{X}} \left\|D_{y,y'}^k A\left(\mathbf{x}\right)\right\|.$$

In typical applications $c_A = O\left(1/n\right)$ (compare the SVM-application in Rivasplata et al. [2018], as derived originally from Bousquet and Elisseeff [2002]).

Consider first the algorithm arising from the Hamiltonian

$$H\left(h, \mathbf{x}\right) = -G\left(h - A\left(\mathbf{x}\right)\right), \tag{6}$$

where $G : \mathbb{R}^d \to [0, \infty)$ is any function with Lipschitz norm $\|G\|_{\text{Lip}}$. One computes $A\left(\mathbf{x}\right)$ and samples $h$ from the stochastic kernel proportional to $\exp\left(-G\left(h - A\left(\mathbf{x}\right)\right)\right)$. By the triangle inequality $H$ satisfies the bounded difference conditions of Theorems 3.4 and 3.5 with $c = \|G\|_{\text{Lip}} c_A$. If every $h \in \mathcal{H}$ (as a function on $\mathcal{X}$) has range in $[0, 1]$, then, although this algorithm is of a completely different nature, we immediately recover the generalization guarantees as for the Gibbs-algorithm with $\beta/n$ replaced by $\|G\|_{\text{Lip}} c_A$. An obvious example is $G\left(h\right) = \|h\|/\sigma$ for $\sigma > 0$.

Another interesting algorithm arises from the Hamiltonian

$$H\left(h, \mathbf{x}\right) = -\frac{\|h - A\left(\mathbf{x}\right)\|^2}{2\sigma^2},$$

for $\sigma > 0$, corresponding to gaussian randomization. For stochastic hypotheses there is an elegant treatment by Rivasplata et al. [2018] using the PAC-Bayesian theorem, and resulting in the bound (with probability at least $1 - \delta$ as $\mathbf{X} \sim \mu^n$)

$$\mathbb{E}_{h \sim Q_{\mathbf{X}}} \left[ kl \left( \hat{L} (h, \mathbf{X}), L(h) \right) \right] \leq \frac{\frac{nc_A^2}{2\sigma^2} \left( 1 + \sqrt{\frac{1}{2} \ln \left( \frac{1}{\delta} \right)} \right)^2 + \ln \left( \frac{2\sqrt{n}}{\delta} \right)}{n}. \tag{7}$$

Since the squared norm is not Lipschitz the previous argument does not work, but with a slight variation of the method we can prove the following result (proof in Section B.1).

**Theorem 4.1.** *Let $\mathcal{H} = \mathbb{R}^d$ with Lebesgue measure $\pi$. Suppose $Q$ has Hamiltonian $H(h, \mathbf{X}) = -\|h - A(\mathbf{X})\|^2 / 2\sigma^2$, where $A$ has stability coefficient $c_A$. Let $\delta > 0$ and assume that $12nc_A^2 \leq \sigma^2$ and that every $h \in \mathcal{H}$ (as a function on $\mathcal{X}$) has range in $[0, 1]$. Denote the variance of $A$ by $\mathcal{V}(A) = \mathbb{E} \left[ \|A(\mathbf{X}) - \mathbb{E}[A(\mathbf{X}')]\|^2 \right]$. Then*

*(i) If $n > 8$ then $\ln \mathbb{E}_{\mathbf{X}} \left[ \mathbb{E}_{h \sim Q_{\mathbf{X}}} \left[ e^{(n/2)kl(\hat{L}(h,\mathbf{X}),L(h))} \right] \right] \leq \frac{3}{\sigma^2} \mathcal{V}(A) + \frac{1}{2} \ln \left( 2\sqrt{n} \right).$*

*(ii) If $n > 8$ then with probability at least $1 - \delta$ as $\mathbf{X} \sim \mu^n$*

$$\mathbb{E}_{h \sim Q_{\mathbf{X}}} \left[ kl \left( \hat{L} (h, \mathbf{X}), L(h) \right) \right] \leq \frac{\frac{6}{\sigma^2} \mathcal{V}(A) + \ln \left( 2\sqrt{n} \right) + 2 \ln (1/\delta)}{n}.$$

*(iii) With probability at least $1 - \delta$ as $\mathbf{X} \sim \mu^n$ and $h \sim Q_{\mathbf{X}}$*

$$\Delta (h, \mathbf{X}) \leq \sqrt{\frac{(3/\sigma^2) \mathcal{V}(A) + \ln (1/\delta)}{n}}.$$

*(iv) Let $v(h)$ be the variance of $h$, defined as in Theorem 3.5. Then with probability at least $1 - \delta$ as $\mathbf{X} \sim \mu^n$ and $h \sim Q_{\mathbf{X}}$*

$$\Delta (h, \mathbf{X}) \leq 2 \sqrt{v(h) \frac{(3/\sigma^2) \mathcal{V}(A) + \ln (1/\delta)}{n}} + \frac{(3/\sigma^2) \mathcal{V}(A) + \ln (1/\delta)}{n}.$$

The expected $kl$-bound (ii) is given only for direct comparison with (7). (iii) and (vi) are stronger, not only by being disintegrated, but also by avoiding the logarithmic dependence on $n$.

In comparison to (7) (ii) has slightly larger constants and we require that $12nc_A^2 \leq \sigma^2$. The latter assumption is mild and holds for sufficiently large $n$ if $nc_A^2 \to 0$ as $n \to \infty$ (in applications of (7) $c_A = O(1/n)$), but $nc_A^2$ may even remain bounded away from zero for $12nc_A^2 \leq \sigma^2$ to hold. On the other hand $\mathcal{V}(A) = \mathbb{E} \left[ \|A(\mathbf{X}) - \mathbb{E}[A(\mathbf{X}')]\|^2 \right]$ is always bounded above by $nc_A^2$ (see the proof of Lemma 6 in Rivasplata et al. [2018]), so we recover (7) from (ii), while our bound can take advantage of benign distributions. In fortunate cases $\mathcal{V}(A)$ can be arbitrarily close to zero, while the $nc_A^2$ in (7) is a consequence of the use of McDiarmid's inequality in the proof, and (7) remains a worst case bound.

The bound (iv) can be inverted as in Corollary 3.7 to give faster rates for small empirical errors, but without the logarithmic dependence in $n$ as in the inverted version of (ii).

## 4.3 PAC-Bayes bounds with data-dependent priors

We quote Theorem 1 (ii) in (Rivasplata et al. [2020]). For the convenience of the reader we give a proof in the appendix (Section B.2).

**Theorem 4.2.** *Let $F : \mathcal{H} \times \mathcal{X}^n \to R$ be measurable. With probability at least $1 - \delta$ in the draw of $\mathbf{X} \sim \mu^n$ we have for all $P \in \mathcal{P}(\mathcal{H})$*

$$\mathbb{E}_{h \sim P} [F(h, \mathbf{X})] \leq KL(P, Q_{\mathbf{X}}) + \ln \mathbb{E}_{\mathbf{X}} \left[ \mathbb{E}_{h \sim Q_{\mathbf{X}}} \left[ e^{F(h, \mathbf{X})} \right] \right] + \ln (1/\delta).$$

By substitution of our bounds on $\ln \mathbb{E}_{\mathbf{X}} \left[ \mathbb{E}_{h \sim Q_{\mathbf{X}}} \left[ e^{F(h, \mathbf{X})} \right] \right]$ we obtain raw forms of PAC-Bayesian bounds with prior $Q_{\mathbf{X}}$ for all the Hamiltonian algorithms considered above. But since the final

form often involves optimizations, some care is needed. In the simplest case let $F(h, \mathbf{X}) = (n/2) kl \left( \hat{L}(h, \mathbf{X}), L(h) \right)$, substitute (i) of Theorem 4.1 above and divide by $n/2$, to prove the following.

**Theorem 4.3.** *Under the conditions of Theorem 4.1 we have with probability at least $1 - \delta$ in $\mathbf{X} \sim \mu^n$ for all $P \in \mathcal{P}(\mathcal{H})$ that*

$$\mathbb{E}_{h \sim P} \left[ kl \left( \hat{L}(h, \mathbf{X}), L(h) \right) \right] \leq \frac{2KL(P, Q_{\mathbf{X}}) + \frac{6}{\sigma^2} \mathcal{V}(A) + \ln(2\sqrt{n}) + 2\ln(1/\delta)}{n}.$$

It applies to the case, when the prior is an isotropic Gaussian, centered on the output of the algorithm $A$, a method related to the methods in Dziugaite and Roy [2018] and Pérez-Ortiz et al. [2021]. Section B.2 sketches how PAC-Bayesian bounds analogous to 4.1 (iii) and (iv) are obtained.

## 5 Conclusion and future directions

The paper presented a method to bound the generalization gap for randomly generated and deterministically executed hypotheses.

By using Marton's coupling method as for example in Paulin [2015], one can prove an analogue to Theorem 3.4 for non-iid data generated by a uniformly ergodic Markov chain.

It also appears possible to apply the method to iterated stochastic algorithms, where the randomization of a stable "microalgorithm" is repeated, as with stochastic gradient Langevin dynamics (SGLD).

An obvious challenge is to give bounds for the Gibbs algorithm in the limit $\beta \to \infty$, or, more generally, in the regime $\beta > n$. It is unlikely that the methods of this paper can be successfully applied to this problem without considerable modifications.

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

# A  Remaining proofs of Section 3

## A.1  Markov's inequality

We use the following consequence of Markov's inequality.

**Lemma A.1.** *For any real random variable $Y$ and $\delta > 0$ we have*

$$\Pr\left\{Y > \ln \mathbb{E}\left[e^Y\right] + \ln\left(1/\delta\right)\right\} \leq \delta.$$

*Proof.* From Markov's inequality $\Pr\left\{e^Y > \mathbb{E}\left[e^Y\right]/\delta\right\} \leq \delta$. Take logarithms. $\qquad\square$

## A.2  Proof of Proposition 3.2

**Lemma A.2.** *(i) Let $\varphi(t) = \left(e^t - t - 1\right)/t^2$ if $t \neq 0$. Then the function $\varphi$ is increasing, and if the random variable $X$ satisfies $\mathbb{E}[X] = 0$ and $X \leq b$ for $b > 0$, then*

$$\mathbb{E}\left[e^X\right] \leq e^{\varphi(b)\mathbb{E}\left[X^2\right]}.$$

*(ii) $\varphi(t) \leq 1/\left(2 - t\right)$ for $0 \leq t < 2$.*

*Proof.* Part (i) is Lemma 2.8 in McDiarmid [1998]. (ii) follows from a term by term comparison of the power series

$$\varphi(t) = \sum_{k=0}^{\infty} \frac{t^k}{(k+2)!} \text{ and } \frac{1}{2-t} = \sum_{k=0}^{\infty} \frac{t^k}{2^{k+1}}.$$

$\qquad\square$

**Proposition A.3** (Restatement of Proposition 3.2)**.** *Let $X, X_1, ..., X_n$ be iid random variables with values in $\mathcal{X}$, $\mathbf{X} = (X_1, ..., X_n)$ and $f : \mathcal{X}^n \to \mathbb{R}$ measurable.*

*(i) If $f$ is such that for all $k \in [n]$, $\mathbf{x} \in \mathcal{X}^n$ we have $\mathbb{E}_X\left[e^{f\left(S_X^k \mathbf{x}\right) - \mathbb{E}_{X'}\left[f\left(S_{X'}^k \mathbf{x}\right)\right]}\right] \leq e^{r^2}$, then $\mathbb{E}\left[e^{f(\mathbf{X}) - \mathbb{E}[f(\mathbf{X}')]}\right] \leq e^{nr^2}$.*

*(ii) If $D_{y,y'}^k f(\mathbf{x}) \leq c$ for all $k \in [n]$, $y, y' \in \mathcal{X}$ and $\mathbf{x} \in \mathcal{X}^n$, then $\mathbb{E}\left[e^{f(\mathbf{X}) - \mathbb{E}[f(\mathbf{X}')]}\right] \leq e^{nc^2/8}$.*

*(iii) If there is $b \in (0, 2)$, such that for all $k \in [n]$ and $\mathbf{x} \in \mathcal{X}^n$ we have $f(\mathbf{x}) - \mathbb{E}_{X' \sim \mu}\left[f\left(S_{X'}^k \mathbf{x}\right)\right] \leq b$, then with $v_k = \sup_{x \in \mathcal{X}^n} \mathbb{E}_{X \sim \mu}\left[\left(f\left(S_X^k \mathbf{x}\right) - \mathbb{E}_{X' \sim \mu}\left[f\left(S_{X'}^k \mathbf{x}\right)\right]\right)^2\right]$*

$$\mathbb{E}_{\mathbf{X}}\left[e^{f(\mathbf{X}) - \mathbb{E}[f(\mathbf{X}')]}\right] \leq \exp\left(\frac{1}{2-b}\sum_{k=1}^{n} v_k\right).$$

*Proof.* (i) For $S \subseteq [n]$ we write $\mathbb{E}_S\left[.\right] = \mathbb{E}\left[.|\left\{X_i\right\}_{i \notin S}\right]$, so $\mathbb{E}_S\left[.\right]$ is integration over all variables in $S$. By independence $\left\{\mathbb{E}_S\left[.\right] : S \subseteq [n]\right\}$ is a set of commuting projections and $\mathbb{E}_{S_1}\left[\mathbb{E}_{S_2}\left[.\right]\right] = \mathbb{E}_{S_1 \cup S_2}\left[.\right]$. $\mathbb{E}_{[k]}$ is expectation in all variables up to $X_k$, and $\mathbb{E}_{\{k\}}$ is expectation only in $X_k$. The assumption therefore reads

$$\mathbb{E}_{\{k\}}\left[e^{f(X) - \mathbb{E}_{\{k\}}[f(X)]}\right] \leq e^{r^2}.$$

Using $\mathbb{E}_{[k-1]}\left[\mathbb{E}_{[k]}\left[f(X)\right]\right] = \mathbb{E}_{[k-1]}\left[\mathbb{E}_{\{k\}}\left[f(X)\right]\right]$. We have the telescopic expansion

$$
\begin{aligned}
f(X) - \mathbb{E}\left[f(X')\right] &= \sum_{k=1}^{n} \mathbb{E}_{[k-1]}\left[f(X)\right] - \mathbb{E}_{[k]}\left[f(X)\right] \\
&= \sum_{k=1}^{n} \mathbb{E}_{[k-1]}\left[f(X) - \mathbb{E}_{\{k\}}\left[f(X)\right]\right],
\end{aligned}
$$

We claim that for all $m$, $0 \leq m \leq n$

$$\mathbb{E}\left[e^{f(X)-\mathbb{E}[f(X')]}\right] \leq e^{mr^2}\mathbb{E}\left[\exp\left(\sum_{k=m+1}^{n}\mathbb{E}_{[k-1]}\left[f(X)-\mathbb{E}_{\{k\}}\left[f(X)\right]\right]\right)\right]$$

from which the proposition follows with $m = n$. Because of above telescopic expansion the claim is true for $m = 0$, and we assume it to hold for $m - 1$. Then

$$\mathbb{E}\left[e^{f(X)-\mathbb{E}[f(X')]}\right]$$

$$\leq e^{(m-1)r^2}\mathbb{E}\left[\exp\left(\sum_{k=m}^{n}\mathbb{E}_{[k-1]}\left[f(X)-\mathbb{E}_{\{k\}}\left[f(X)\right]\right]\right)\right]$$

$$= e^{(m-1)r^2}\mathbb{E}\left[\exp\left(\mathbb{E}_{[m-1]}\left[f(X)-\mathbb{E}_{\{m\}}\left[f(X)\right] + \sum_{k=m+1}^{n}\mathbb{E}_{[k-1]}\left[f(X)-\mathbb{E}_{\{k\}}\left[f(X)\right]\right]\right]\right)\right]$$

because the later terms depend only on the variables $X_{m+1}, ..., X_n$. By Jensen's inequality the last expression is bounded by

$$e^{(m-1)r^2}\mathbb{E}\left[\exp\left(f(X)-\mathbb{E}_{\{m\}}\left[f(X)\right] + \sum_{k=m+1}^{n}\mathbb{E}_{[k-1]}\left[f(X)-\mathbb{E}_{\{k\}}\left[f(X)\right]\right]\right)\right]$$

$$= e^{(m-1)r^2}\mathbb{E}\left[e^{f(X)-\mathbb{E}_{\{m\}}[f(X)]}\exp\left(\sum_{k=m+1}^{n}\mathbb{E}_{[k-1]}\left[f(X)-\mathbb{E}_{\{k\}}\left[f(X)\right]\right]\right)\right]$$

$$= e^{(m-1)r^2}\mathbb{E}\left[\mathbb{E}_{\{m\}}\left[e^{f(X)-\mathbb{E}_{\{m\}}[f(X)]}\right]\exp\left(\sum_{k=m+1}^{n}\mathbb{E}_{[k-1]}\left[f(X)-\mathbb{E}_{\{k\}}\left[f(X)\right]\right]\right)\right],$$

again because the later terms do not depend on $X_m$, and by assumption the last expression is bounded by

$$e^{mr^2}\mathbb{E}\left[\exp\left(\sum_{k=m+1}^{n}\mathbb{E}_{[k-1]}\left[f(X)-\mathbb{E}_{\{k\}}\left[f(X)\right]\right]\right)\right],$$

which completes the induction and the proof of (i).

(ii) follows from (i) and Hoeffding's lemma (Lemma 2.2 in Boucheron et al. [2013]) which says that

$$\mathbb{E}_{X\sim\mu}\left[e^{f\left(S_X^k\mathbf{x}\right)-\mathbb{E}_{X'\sim\mu}\left[f\left(S_{X'}^k\mathbf{x}\right)\right]}\right] \leq e^{\frac{c^2}{8}},$$

if $f\left(S_y^k\mathbf{x}\right)$ as a function of $y$ has range in a set of diameter $c$.

(iii) Follows from (i) and Lemma A.2 since

$$\mathbb{E}_{X\sim\mu}\left[e^{f\left(S_X^k\mathbf{x}\right)-\mathbb{E}_{X'\sim\mu}\left[f\left(S_{X'}^k\mathbf{x}\right)\right]}\right] \leq e^{\varphi(b)v_k}.$$

$\square$

## A.3  Proof of Corollary 3.7

**Lemma A.4.** *Let $L, \hat{L}, A \geq 0$ and assume that*

$$L \leq \hat{L} + 2\sqrt{L}\sqrt{A} + A$$

*Then $L \leq \hat{L} + 2\sqrt{\hat{L}A} + 5A$*

*Proof.*

$$
\begin{aligned}
L &\leq \hat{L} + 2\sqrt{L}\sqrt{A} + A \iff L - 2\sqrt{L}\sqrt{A} + A \leq \hat{L} + 2A \\
&\iff \left(\sqrt{L} - \sqrt{A}\right)^2 \leq \hat{L} + 2A \implies \sqrt{L} \leq \sqrt{\hat{L} + 2A} + \sqrt{A} \\
&\iff L \leq \left(\sqrt{\hat{L} + 2A} + \sqrt{A}\right)^2 \leq \hat{L} + 2\sqrt{\hat{L}A} + \left(3 + \sqrt{2}\right)A.
\end{aligned}
$$

The lemma follows from $3 + \sqrt{2} \leq 5$. $\square$

To get Corollary 3.7 apply this to Theorem 3.5.

## A.4 Subgaussian hypotheses and proof of Theorem 3.8

**Lemma A.5.** *(From Buldygin and Kozachenko [1980]) (i) if $Y$ is $\sigma$-subgaussian then $\mathbb{E}\left[(Y - \mathbb{E}[Y])^2\right] \leq \sigma^2$. (ii) if $Y_1$ and $Y_2$ are $\sigma_1$ - and $\sigma_2$-subgaussian respectively, the $Y_1 + Y_2$ is $\sigma_1 + \sigma_2$-subgaussian.*

The next lemma shows that the log-partition function $\ln Z(\mathbf{X})$ is exponentially concentrated, whenever the Hamiltonian $H(h, \mathbf{X})$ is subgaussian uniformly in $h$.

**Lemma A.6.** *Let $p \geq 1$ and $H(h, \mathbf{X})$ be $\sigma$-subgaussian for every $h \in \mathcal{H}$ and*

$$Z(\mathbf{x}) = \int_{\mathcal{H}} e^{H(h, \mathbf{x})} d\pi(h).$$

*(i) Then $\ln \mathbb{E}\left[e^{p\left(-\ln Z(\mathbf{X}) + \mathbb{E}\left[\ln Z(\mathbf{X}')\right]\right)}\right] \leq p^2 \sigma^2$.*

*(ii) If $f : \mathcal{X}^n \to \mathbb{R}$ is $\rho$-subgaussian then*

$$\ln \mathbb{E}\left[e^{p\left(f(\mathbf{X}) - \mathbb{E}[f(\mathbf{X}')] - \ln Z(\mathbf{X}) + \mathbb{E}[\ln Z(\mathbf{X}')]\right)}\right] \leq p^2 (\rho + \sigma)^2.$$

Since the inequalities are given only for $p \geq 1$ they do not quite imply that $\ln Z$ itself is subgaussian.

*Proof.* We only need to prove (ii), which implies (i) by setting $f \equiv 0$. By Jensen's inequality

$$\mathbb{E}_{\mathbf{X}}\left[e^{p\left(f(\mathbf{X}) - \mathbb{E}[f(\mathbf{X}')] - \ln Z(\mathbf{X}) + \mathbb{E}[\ln Z(\mathbf{X}')]\right)}\right] \leq \mathbb{E}_{\mathbf{X}\mathbf{X}'}\left[e^{p\left(f(\mathbf{X}) - f(\mathbf{X}') - \ln Z(\mathbf{X}) + \ln Z(\mathbf{X}')\right)}\right]$$

$$= \mathbb{E}_{\mathbf{X}}\left[e^{pf(\mathbf{X})} Z(\mathbf{X})^{-p}\right] \mathbb{E}_{\mathbf{X}}\left[e^{-pf(\mathbf{X})} Z(\mathbf{X})^{p}\right].$$

Define a probability measure $\nu$ on $\mathcal{H}$ by $\nu(A) = Z_\nu^{-1} \int_A e^{\mathbb{E}[H(h, \mathbf{X})]} d\pi(h)$ for $A \subseteq \mathcal{H}$ measurable.with $Z_\nu = \int_{\mathcal{H}} e^{\mathbb{E}[H(h, \mathbf{X})]} d\pi(h)$. Then

$$Z(\mathbf{X})^{-p} = \left(\mathbb{E}_{h \sim \nu}\left[e^{H(h, \mathbf{X}) - \mathbb{E}[H(h, \mathbf{X}')]}\right]\right)^{-p} Z_\nu^p \leq \mathbb{E}_{h \sim \nu}\left[e^{p\left(\mathbb{E}[H(h, \mathbf{X}')] - H(h, \mathbf{X})\right)}\right] Z_\nu^p,$$

by Jensen's inequality, since $t \mapsto t^{-1}$ is convex. Similarly

$$Z(\mathbf{X})^p \leq \mathbb{E}_{h \sim \nu}\left[e^{p\left(H(h, \mathbf{X}) - \mathbb{E}[H(h, \mathbf{X}')]\right)}\right] Z_\nu^{-p}.$$

Thus the above inequality can be written

$$\mathbb{E}_{\mathbf{X}}\left[e^{p\left(f(\mathbf{X}) - \mathbb{E}[f(\mathbf{X}')] - \ln Z(\mathbf{X}) + \mathbb{E}[\ln Z(\mathbf{X}')]\right)}\right] \leq \mathbb{E}_{\mathbf{X}}\left[\mathbb{E}_{h \sim \nu}\left[e^{p\left(f(\mathbf{X}) + \mathbb{E}[H(h, \mathbf{X}')] - H(h, \mathbf{X})\right)}\right]\right]$$

$$\times \mathbb{E}_{\mathbf{X}}\left[\mathbb{E}_{h \sim \nu}\left[e^{p\left(-f(\mathbf{X}) + H(h, \mathbf{X}) - \mathbb{E}[H(h, \mathbf{X}')]\right)}\right]\right].$$

The first factor can be bounded by

$$\mathbb{E}_{h \sim \nu}\left[\mathbb{E}_{\mathbf{X}}\left[e^{p\left(f(\mathbf{X}) + \mathbb{E}[H(h, \mathbf{X}')] - H(h, \mathbf{X})\right)}\right]\right] \leq e^{p\mathbb{E}[f(\mathbf{X}')]} e^{\frac{p^2(\rho + \sigma)^2}{2}}$$

by the subgaussian assumptions for $f$ and $H$ and Lemma A.5 (ii), and similarly the second factor is bounded by

$$e^{-p\mathbb{E}[f(\mathbf{X}')]} e^{\frac{p^2(\rho + \sigma)^2}{2}}.$$

Putting the two bounds together completes the proof. $\qquad \square$

**Theorem A.7** (Restatement of Theorem 3.8). *Let $Q$ have Hamiltonian $H$ and assume that $\forall h \in \mathcal{H}$ there is $\rho(h) > 0$ such that*

$$\forall \lambda \in \mathbb{R}, \, \mathbb{E}\left[e^{\lambda\left(h(X) - \mathbb{E}[h(X')]\right)}\right] \leq e^{\frac{\lambda^2 \rho(h)^2}{2}},$$

*Let $\rho = \sup_{h \in \mathcal{H}} \rho(h)$ and suppose that $\forall \lambda \in \mathbb{R}, k \in [n], h \in \mathcal{H}$*

$$\mathbb{E}\left[e^{\lambda\left(H\left(h, S_X^k \mathbf{x}\right) - \mathbb{E}\left[H\left(h, S_{X'}^k \mathbf{x}\right)\right]\right)}\right] \leq e^{\frac{\lambda^2 \sigma^2}{2}}.$$

*(i) Then for any $h \in \mathcal{H}$, $\lambda > 0$*

$$\ln \mathbb{E}_{\mathbf{X} \sim \mu^n} \mathbb{E}_{h \sim Q_{\mathbf{X}}} \left[e^{\lambda \Delta}\right] \leq \psi_{\lambda\Delta}(h) \leq \frac{\left(\lambda\rho(h)/\sqrt{n} + 2\sqrt{n}\sigma\right)^2}{2},$$

*and with probability at least $1 - \delta$ we have as $\mathbf{X} \sim \mu^n$ and $h \sim Q_{\mathbf{X}}$ that*

$$\Delta(h, \mathbf{X}) \leq \rho\left(2\sigma + \sqrt{\frac{2\ln(1/\delta)}{n}}\right).$$

*(ii) With probability at least $1 - \delta$ we have as $\mathbf{X} \sim \mu^n$ and $h \sim Q_{\mathbf{X}}$ that*

$$\Delta(h, \mathbf{X}) \leq \rho(h)\left(\sqrt{32}\sigma + \sqrt{\frac{4\ln(1/\delta)}{n}}\right).$$

*Proof.* Let $h \in \mathcal{H}$ be any fixed hypothesis. By assumption and Proposition 3.2 (i) $H(h, \mathbf{X})$ is $\sqrt{n}\sigma$-subgaussian and $\lambda\Delta(h, \mathbf{X})$ is $\lambda\rho(h)/\sqrt{n}$-subgaussian.

Using the previous lemma (ii) with $p = 1$ and $f(\mathbf{X}) = \lambda\Delta(h, \mathbf{X}) + H(h, \mathbf{X}) - \mathbb{E}[H(h, \mathbf{X}')]$, which is centered and $\lambda\rho(h)/\sqrt{n} + \sqrt{n}\sigma$-subgaussian, we get that

$$\begin{aligned}
\psi_{\lambda\Delta}(h) &= \ln \mathbb{E}_{\mathbf{X}}\left[e^{\lambda\Delta(h,\mathbf{X}) + H_Q(h,\mathbf{X}) - \mathbb{E}\left[H_Q\left(h,\mathbf{X}'\right)\right]}\right] \\
&= \ln \mathbb{E}\left[e^{f(\mathbf{X}) - \ln Z(\mathbf{X}) + \mathbb{E}\left[\ln Z\left(\mathbf{X}'\right)\right]}\right] \\
&\leq \frac{\left(\lambda\rho(h)/\sqrt{n} + 2\sqrt{n}\sigma\right)^2}{2}.
\end{aligned}$$

With $\rho(g) \leq \rho$ we get from Proposition 3.1 that with probability at least $1 - \delta$

$$\begin{aligned}
\Delta(h, \mathbf{X}) &\leq \frac{\left(\lambda\rho/\sqrt{n} + 2\sqrt{n}\sigma\right)^2}{2\lambda} + \frac{\ln(1/\delta)}{\lambda} \\
&= \frac{\lambda\rho^2}{2n} + \frac{2n\sigma^2 + \ln(1/\delta)}{\lambda} + 2\rho\sigma
\end{aligned}$$

The optimal choice of $\lambda$ and subadditivity of $t \to \sqrt{t}$ give

$$\Delta(h, \mathbf{X}) \leq \rho\left(2\sigma + \sqrt{\frac{2\ln(1/\delta)}{n}}\right).$$

(ii) We proceed as in the proof of Theorem 3.5. For $H_Q(h, \mathbf{X}) = H(h, \mathbf{X}) - \ln Z(\mathbf{X})$ the previous lemma yields with $f(\mathbf{X}) = H(g, \mathbf{X}) - \mathbb{E}[H(g, \mathbf{X}')]$ and $p = 2$ that

$$\mathbb{E}_{\mathbf{X}}\left[e^{2\left(H_Q(h,\mathbf{X}) - \mathbb{E}\left[H_Q\left(h,\mathbf{X}'\right)\right]\right)}\right] \leq e^{8n\sigma^2}.$$

Also

$$\mathbb{E}_{\mathbf{X}}\left[e^{2\lambda\Delta(h,\mathbf{X})}\right] \leq e^{2\lambda^2 \rho(h)^2/n}.$$

Now define

$$\lambda\left(h\right)=\sqrt{\left(\frac{n}{\rho\left(h\right)^2}\right)\left(8n\sigma^2+\ln\left(1/\delta\right)\right)}$$

and $F\left(h,\mathbf{X}\right)=\lambda\left(h\right)\Delta\left(h,\mathbf{X}\right)-\lambda\left(h\right)^2\rho\left(h\right)^2/n$. Then with Cauchy-Schwarz

$$
\begin{aligned}
\psi_F\left(h\right) &= \ln\mathbb{E}\left[e^{F\left(h,\mathbf{X}\right)+H_Q\left(h,\mathbf{X}\right)-\mathbb{E}\left[H_Q\left(h,\mathbf{X}'\right)\right]}\right]\\
&\leq \ln\left(\left(\mathbb{E}_\mathbf{X}\left[e^{2\lambda\Delta\left(h,\mathbf{X}\right)}\right]\right)^{1/2}e^{-\lambda\left(h\right)^2\rho\left(h\right)^2/n}\mathbb{E}_\mathbf{X}\left[e^{2\left(H_Q\left(h,\mathbf{X}\right)-\mathbb{E}\left[H_Q\left(h,\mathbf{X}'\right)\right]\right)}\right]^{1/2}\right)\\
&\leq 8n\sigma^2.
\end{aligned}
$$

Proposition 3.1, substitution of $\lambda\left(h\right)$ and subadditivity of $t\to\sqrt{t}$ then give

$$
\begin{aligned}
\Delta\left(h,\mathbf{X}\right) &\leq \frac{\lambda\left(h\right)\rho\left(h\right)^2}{n}+\frac{8n\sigma^2+\ln\left(1/\delta\right)}{\lambda\left(h\right)}\\
&= \sqrt{\frac{\rho\left(h\right)^2}{n}\left(32n\sigma^2+4\ln\left(1/\delta\right)\right)}\\
&= \rho\left(h\right)\left(\sqrt{32}\sigma+\sqrt{\frac{4\ln\left(1/\delta\right)}{n}}\right).
\end{aligned}
$$

$\square$

# B   Remaining proofs for Section 4

## B.1   Proof of Theorem 4.1

We need the following Lemma.

**Lemma B.1.** *Let $w,v\in\mathbb{R}^d$ and $\lambda\in[1,\infty)$ then*

$$\mathbb{E}_{x\sim\mathcal{N}\left(w,\sigma^2 I\right)}\left[e^{\left(\frac{-\lambda}{2\sigma^2}\right)\left(\|x-v\|^2-\|x-w\|^2\right)}\right]=e^{\left(\frac{2\lambda^2-\lambda}{2\sigma^2}\right)\|v-w\|^2}.$$

*Proof.* We can absorb $\sqrt{2}\sigma$ in the definition of the norm. Then by translation

$$
\begin{aligned}
\mathbb{E}_{x\sim\mathcal{N}\left(w,I\right)}\left[e^{-\lambda\left(\|x-v\|^2-\|x-w\|^2\right)}\right] &= \mathbb{E}_{x\sim\mathcal{N}\left(0,I\right)}\left[e^{-\lambda\left(\|x-(v-w)\|^2-\|x\|^2\right)}\right]\\
&= e^{-\lambda\|v-w\|^2}\mathbb{E}_{x\sim\mathcal{N}\left(0,I\right)}\left[e^{2\lambda\langle x,v-w\rangle}\right].
\end{aligned}
$$

Rotating $v-w$ to $\|v-w\|\,e_1$, where $e_1$ is the first basis vector, and using independence of the components gives

$$
\begin{aligned}
\mathbb{E}_{x\sim\mathcal{N}\left(0,I\right)}\left[e^{2\lambda\langle x,v-w\rangle}\right] &= \mathbb{E}_{x\sim\mathcal{N}\left(0,I\right)}\left[e^{2\lambda\|v-w\|\langle x,e_1\rangle}\right]=\frac{1}{\sqrt{2\pi}}\int_{-\infty}^{\infty}e^{2\lambda\|v-w\|t-t^2/2}dt\\
&= \frac{e^{2\lambda^2\|v-w\|^2}}{\sqrt{2\pi}}\int_{-\infty}^{\infty}e^{-\left(\frac{2\lambda\|v-w\|-t}{\sqrt{2}}\right)^2}dt=e^{2\lambda^2\|v-w\|^2}.
\end{aligned}
$$

Combination with the previous identity and taking $\sqrt{2}\sigma$ back out of the norm completes the proof.   $\square$

Our proof of Theorem 4.1 uses the following exponential concentration inequality, implicit in the proof of Theorem 13 in (Maurer [2006]) and in the proof of Theorem 6.19 in (Boucheron et al. [2013])

**Theorem B.2.** *Let $f:\mathcal{X}^n\to\mathbb{R}$ and define an operator $D^2$ by*

$$\left(D^2 f\right)\left(\mathbf{x}\right)=\sum_{k=1}^{n}\left(f\left(\mathbf{x}\right)-\inf_{y\in\mathcal{X}}f\left(S_y^k\mathbf{x}\right)\right)^2.$$

*If for some $a > 0$ and all $\mathbf{x} \in \mathcal{X}^n$, $D^2 f(\mathbf{x}) \leq a f(\mathbf{x})$, then for $\lambda \in (0, 2/a)$*

$$\ln \mathbb{E}\left[e^{\lambda(f(\mathbf{X}) - \mathbb{E}[f(\mathbf{X}')])}\right] \leq \frac{\lambda^2 a \mathbb{E}[f(\mathbf{X})]}{2 - a\lambda} \quad \text{or equivalently } \ln \mathbb{E}\left[e^{\lambda f(\mathbf{X})}\right] \leq \frac{2\lambda \mathbb{E}[f(\mathbf{X})]}{2 - a\lambda}.$$

**Theorem B.3** (Restatement of Theorem 4.1). *Let $\mathcal{H} = \mathbb{R}^d$ with Lebesgue measure $\pi$. Suppose $Q$ has Hamiltonian $H(h, \mathbf{X}) = -\|h - A(\mathbf{X})\|^2 / 2\sigma^2$, where $A$ has stability coefficient $c_A$. Let $\delta > 0$ and assume that $12nc_A^2 \leq \sigma^2$ and that every $h \in \mathcal{H}$ (as a function on $\mathcal{X}$) has range in $[0, 1]$. Denote the variance of $A$ by $\mathcal{V}(A) = \mathbb{E}\left[\|A(\mathbf{X}) - \mathbb{E}[A(\mathbf{X}')]\|^2\right]$. Then*

*(i) If $n > 8$ then $\ln \mathbb{E}_{\mathbf{X}}\left[\mathbb{E}_{h \sim Q_{\mathbf{X}}}\left[e^{(n/2)kl\left(\hat{L}(h,\mathbf{X}), L(h)\right)}\right]\right] \leq \frac{3}{\sigma^2} \mathcal{V}(A) + \frac{1}{2}\ln(2\sqrt{n})$.*

*(ii) If $n > 8$ then with probability at least $1 - \delta$ as $\mathbf{X} \sim \mu^n$*

$$\mathbb{E}_{h \sim Q_{\mathbf{X}}}\left[kl\left(\hat{L}(h, \mathbf{X}), L(h)\right)\right] \leq \frac{\frac{6}{\sigma^2}\mathcal{V}(A) + \ln(2\sqrt{n}) + 2\ln(1/\delta)}{n}.$$

*(iii) With probability at least $1 - \delta$ as $\mathbf{X} \sim \mu^n$ and $h \sim Q_{\mathbf{X}}$*

$$\Delta(h, \mathbf{X}) \leq \sqrt{\frac{(3/\sigma^2)\mathcal{V}(A) + \ln(1/\delta)}{n}}.$$

*(iv) Let $v(h)$ be the variance of $h$, defined as in Theorem 3.5. Then with probability at least $1 - \delta$ as $\mathbf{X} \sim \mu^n$ and $h \sim Q_{\mathbf{X}}$*

$$\Delta(h, \mathbf{X}) \leq 2\sqrt{v(h)\frac{(3/\sigma^2)\mathcal{V}(A) + \ln(1/\delta)}{n}} + \frac{(3/\sigma^2)\mathcal{V}(A) + \ln(1/\delta)}{n}.$$

*Proof of Theorem 4.1.* All Gaussians with covariance $\sigma^2 I$ have the same normalizing factors, so the partition function for $H(h, \mathbf{x}) = -\|h - A(\mathbf{x})\|^2 / (2\sigma^2)$ is also the normalizing factor of $\mathcal{N}(\mathbb{E}[A(\mathbf{X})], \sigma^2 I)$, whence, using Cauchy-Schwarz,

$$\ln \mathbb{E}_{\mathbf{X}}\left[\mathbb{E}_{h \sim Q_{\mathbf{X}}}\left[e^{F(h,\mathbf{X})}\right]\right] = \ln \mathbb{E}_{\mathbf{X}}\left[\int_{\mathcal{H}} e^{F(h,\mathbf{X}) + H_Q(h,\mathbf{X})} d\pi(h)\right]$$

$$= \ln \mathbb{E}_{\mathbf{X}}\left[\mathbb{E}_{h \sim \mathcal{N}(\mathbb{E}[A(\mathbf{X})], \sigma^2 I)}\left[e^{F(h,\mathbf{X}) - \frac{\|h - A(\mathbf{X})\|^2}{2\sigma^2} + \frac{\|h - \mathbb{E}[A(\mathbf{X})]\|^2}{2\sigma^2}}\right]\right]$$

$$\leq \frac{1}{2}\sup_{h \in \mathcal{H}} \ln \mathbb{E}_{\mathbf{X}}\left[e^{2F(h,\mathbf{X})}\right] + \frac{1}{2}\ln \mathbb{E}_{\mathbf{X}}\left[\mathbb{E}_{h \sim \mathcal{N}(\mathbb{E}[A(\mathbf{X})], \sigma^2 I)}\left[e^{-2\left(\frac{\|h - A(\mathbf{X})\|^2}{2\sigma^2} - \frac{\|h - \mathbb{E}[A(\mathbf{X})]\|^2}{2\sigma^2}\right)}\right]\right]$$

$$=: C + B.$$

The bound on $C$ depends on the respective choice of $F$ and will be treated below. Using Lemma B.1 the second term is equal to

$$B = \frac{1}{2}\ln \mathbb{E}_{\mathbf{X}}\left[e^{\frac{3}{\sigma^2}\|A(\mathbf{X}) - \mathbb{E}[A(\mathbf{X})]\|^2}\right].$$

To apply Theorem B.2 to $f(\mathbf{x}) = \|A(\mathbf{x}) - \mathbb{E}[A(\mathbf{X})]\|^2$ we fix $\mathbf{x} \in \mathcal{X}^n$ and $k \in [n]$, and let $y \in \mathcal{X}$ be a minimizer of $\|A(S_y^k \mathbf{x}) - \mathbb{E}[A(\mathbf{X})]\|^2$. Then

$$\left(f(\mathbf{x}) - \inf_{y \in \mathcal{X}} f(S_y^k \mathbf{x})\right)^2 = \left(\|A(\mathbf{x}) - \mathbb{E}[A(\mathbf{X})]\|^2 - \|A(S_y^k \mathbf{x}) - \mathbb{E}[A(\mathbf{X})]\|^2\right)^2$$

$$= \left\langle A(\mathbf{x}) - A(S_y^k \mathbf{x}), A(\mathbf{x}) - \mathbb{E}[A(\mathbf{X})] + A(S_y^k \mathbf{x}) - \mathbb{E}[A(\mathbf{X})]\right\rangle^2$$

$$\leq \|A(\mathbf{x}) - A(S_y^k \mathbf{x})\|^2 \left(\|A(\mathbf{x}) - \mathbb{E}[A(\mathbf{X})]\| + \|A(S_y^k \mathbf{x}) - \mathbb{E}[A(\mathbf{X})]\|\right)^2 \leq 4c_A^2 f(\mathbf{x}).$$

Summing over $k$ we get $D^2 f(\mathbf{x}) \leq 4nc_A^2 f(\mathbf{x})$. Since $12nc_A^2/\sigma^2 \leq 1 < 2$ we can apply Theorem B.2 $a = 4nc_A^2$ and $\lambda = 3/\sigma^2$ to obtain

$$
\begin{aligned}
B &= \frac{1}{2} \ln \mathbb{E}\left[e^{(3/\sigma^2)\|A(\mathbf{X}) - \mathbb{E}[A(\mathbf{X}')]\|^2}\right] \leq \frac{(3/\sigma^2)\mathbb{E}\left[\|A(\mathbf{X}) - \mathbb{E}[A(\mathbf{X}')]\|^2\right]}{2 - 12nc_A^2/\sigma^2} \\
&\leq \frac{3}{\sigma^2}\mathbb{E}\left[\|A(\mathbf{X}) - \mathbb{E}[A(\mathbf{X}')]\|^2\right] = \frac{3}{\sigma^2}\mathcal{V}(A).
\end{aligned}
$$

$\square$

(i) Let $F(h, \mathbf{X}) = (n/2) kl\left(\hat{L}(h, \mathbf{X}), L(h)\right)$. Then using (Maurer [2004]) $C = \sup_h \ln \mathbb{E}_{\mathbf{X}}\left[e^{2F(h,\mathbf{X})}\right]/2 \leq (1/2)\ln(2\sqrt{n})$, so from the above

$$
\ln \mathbb{E}_{\mathbf{X}}\left[\mathbb{E}_{h \sim Q_{\mathbf{X}}}\left[e^{(n/2)kl\left(\hat{L}(h,\mathbf{X}),L(h)\right)}\right]\right] \leq \frac{3}{\sigma^2}\mathcal{V}(A) + \frac{1}{2}\ln(2\sqrt{n}),
$$

which is (i). By Jensen's inequality

$$
\ln \mathbb{E}_{\mathbf{X}}\left[e^{(n/2)\mathbb{E}_{h\sim Q_{\mathbf{X}}}\left[kl\left(\hat{L}(h,\mathbf{X}),L(h)\right)\right]}\right] \leq \ln \mathbb{E}_{\mathbf{X}}\left[\mathbb{E}_{h\sim Q_{\mathbf{X}}}\left[e^{(n/2)kl\left(\hat{L}(h,\mathbf{X}),L(h)\right)}\right]\right],
$$

so part (ii) then follows from Markov's inequality and division by $n/2$.

(iii) Let $F(h, \mathbf{X}) = \lambda\Delta(h, \mathbf{X})$ for $\lambda > 0$. Using Proposition 3.2 (ii) we get for all $h \in \mathcal{H}$ that $C = (1/2)\ln \mathbb{E}_{\mathbf{X}}\left[e^{2F(h,\mathbf{X})}\right] \leq \lambda^2/(4n)$, so

$$
\ln \mathbb{E}_{\mathbf{X}}\left[\mathbb{E}_{h\sim Q_{\mathbf{X}}}\left[e^{\lambda\Delta(h,\mathbf{X})}\right]\right] = \frac{\lambda^2}{4n} + \frac{3}{\sigma^2}\mathcal{V}(A). \tag{8}
$$

Markov's inequality gives with probability at least $1 - \delta$ as $\mathbf{X} \sim \mu^n$ and $h \sim Q_{\mathbf{X}}$ that

$$
\Delta(h, \mathbf{X}) \leq \frac{\lambda}{4n} + \frac{\frac{3}{\sigma^2}\mathcal{V}(A) + \ln(1/\delta)}{\lambda}.
$$

Optimization of $\lambda$ gives (iii).

(iv) We proceed as in the proof of Theorem 3.5. Let

$$
\lambda(h) = \frac{\sqrt{\frac{3}{\sigma^2}\mathcal{V}(A) + \ln(1/\delta)}}{(1/n)\sqrt{\frac{3}{\sigma^2}\mathcal{V}(A) + \ln(1/\delta)} + \sqrt{\frac{v(h)}{n}}}
$$

and set

$$
F_\lambda(h, \mathbf{X}) = \lambda(h)\Delta(h, \mathbf{X}) - \frac{\lambda(h)^2}{1 - \lambda(h)/n}\frac{v(h)}{n}.
$$

By Lemma 3.6 $2C = \ln \mathbb{E}_{\mathbf{X}}\left[e^{2F_\lambda(h,\mathbf{X})}\right] \leq 0$, so

$$
\ln \mathbb{E}_{\mathbf{X}}\left[\mathbb{E}_{h\sim Q_{\mathbf{X}}}\left[e^{F_\lambda(h,\mathbf{X})}\right]\right] \leq \frac{3}{\sigma^2}\mathcal{V}(A),
$$

and with probability at least $1 - \delta$ as as $\mathbf{X} \sim \mu^n$ and $h \sim Q_{\mathbf{X}}$

$$
\Pr\left\{\Delta(h, \mathbf{X}) > \frac{\lambda(h)}{1 - \lambda(h)/n}\frac{v(h)}{n} + \frac{\frac{3}{\sigma^2}\mathcal{V}(A) + \ln(1/\delta)}{\lambda(h)}\right\} < \delta.
$$

Inserting the definition of $\lambda(h)$ completes the proof.

## B.2 PAC-Bayes bounds with data-dependent priors

**Theorem B.4** (Restatement of Theorem 4.2). *Let $F : \mathcal{H} \times \mathcal{X}^n \to R$ be measurable. With probability at least $1 - \delta$ in the draw of $\mathbf{X} \sim \mu^n$ we have for all $P \in \mathcal{P}(\mathcal{H})$*

$$
\mathbb{E}_{h\sim P}[F(h, \mathbf{X})] \leq KL(P, Q_{\mathbf{X}}) + \ln \mathbb{E}_{\mathbf{X}}\left[\mathbb{E}_{h\sim Q_{\mathbf{X}}}\left[e^{F(h,\mathbf{X})}\right]\right] + \ln(1/\delta).
$$

*Proof.* Let $P \in \mathcal{P}_1(\mathcal{H})$ be arbitrary. Then

$$
\begin{aligned}
\mathbb{E}_{h \sim P}\left[F\left(h, \mathbf{X}\right)\right] - KL\left(P, Q_{\mathbf{X}}\right) &= \ln \exp\left(\mathbb{E}_{h \sim P}\left[F\left(h, \mathbf{X}\right) - \ln\left(dP/dQ_{\mathbf{X}}\left(h\right)\right)\right]\right) \\
&\leq \ln \mathbb{E}_{h \sim P}\left[\exp\left(F\left(h, \mathbf{X}\right) - \ln\left(dP/dQ_{\mathbf{X}}\left(h\right)\right)\right)\right] \\
&= \ln \mathbb{E}_{h \sim P}\left[e^{F(h, \mathbf{X})}(dP/dQ_{\mathbf{X}}\left(h\right))^{-1}\right] \\
&= \ln \mathbb{E}_{h \sim Q_{\mathbf{X}}}\left[e^{F(h, \mathbf{X})}\right].
\end{aligned}
$$

So we only need to bound the last expression, which is independent of $P$. But by Markov's inequality (Lemma A.1) with probability at least $1 - \delta$ in $\mathbf{X} \sim \mu^n$

$$
\begin{aligned}
\ln \mathbb{E}_{h \sim Q_{\mathbf{X}}}\left[e^{F(h, \mathbf{X})}\right] &\leq \ln \mathbb{E}_{\mathbf{X}}\left[\exp\left(\ln \mathbb{E}_{h \sim Q_{\mathbf{X}}}\left[e^{F(h, \mathbf{X})}\right]\right)\right] + \ln\left(1/\delta\right) \\
&= \ln \mathbb{E}_{\mathbf{X}}\left[\mathbb{E}_{h \sim Q_{\mathbf{X}}}\left[e^{F(h, \mathbf{X})}\right]\right] + \ln\left(1/\delta\right).
\end{aligned}
$$

$\square$

We close with a method to deal with the problem of parameter optimization in the derivation of PAC-Bayesian bounds from our results. We apply it to the case of Gaussian priors centered on the output of stable algorithms, but analogous results can be equally derived for Hamiltonians with bounded differences or sub-gaussian Hamiltonians. Our first result is a PAC-Bayesian bound analogous to part (iii) of Theorem 4.1.

**Theorem B.5.** *Under the conditions of Theorem 4.1 we have with probability at least $1 - \delta$ in $\mathbf{X} \sim \mu^n$ for all $P \in \mathcal{P}(\mathcal{H})$ that*

$$
\mathbb{E}_{h \sim P}\left[\Delta\left(h, \mathbf{X}\right)\right] > \sqrt{\frac{\frac{3}{\sigma^2}\mathcal{V}\left(A\right) + 2KL\left(P, Q_{\mathbf{X}}\right) + \ln\left(2KL\left(P, Q_{\mathbf{X}}\right)/\delta\right)}{n}}.
$$

To prove this we first establish the following intermediate result.

**Proposition B.6.** *Under the conditions of Theorem 4.1 let $K > 0$ and $\delta > 0$. Then with probability at least $1 - \delta$ as $\mathbf{X} \sim \mu^n$ we have for any $P \in \mathcal{P}(\mathcal{H})$ with $KL\left(P, Q_{\mathbf{X}}\right) \leq K$ that*

$$
\mathbb{E}_{h \sim P}\left[\Delta\left(h, \mathbf{X}\right)\right] \leq \sqrt{\frac{\frac{3}{\sigma^2}\mathcal{V}\left(A\right) + K + \ln\left(1/\delta\right)}{n}}
$$

*Proof.* If $KL\left(P, Q_{\mathbf{X}}\right) \leq K$ we get from Theorem 4.2 and inequality (8) with probability at least $1 - \delta$ in $\mathbf{X} \sim \mu^n$

$$
\begin{aligned}
\mathbb{E}_{h \sim P}\left[\lambda \Delta\left(h, \mathbf{X}\right)\right] - K &\leq \mathbb{E}_{h \sim P}\left[\lambda \Delta\left(h, \mathbf{X}\right)\right] - KL\left(P, Q_{\mathbf{X}}\right) \\
&\leq \ln \mathbb{E}_{\mathbf{X}}\left[\mathbb{E}_{h \sim Q_{\mathbf{X}}}\left[e^{\lambda \Delta(h, \mathbf{X})}\right]\right] + \ln\left(1/\delta\right) \\
&\leq \frac{\lambda^2}{4n} + \frac{3}{\sigma^2}\mathcal{V}\left(A\right) + \ln\left(1/\delta\right).
\end{aligned}
$$

Bring $K$ to the other side, divide by $\lambda$ and and optimize $\lambda$ to complete the proof. $\square$

To get rid of $K$ we use a model-selection lemma from Anthony and Bartlett [1999].

**Lemma B.7.** *(Lemma 15.6 in Anthony and Bartlett [1999]) Suppose* $\Pr$ *is a probability distribution and*

$$
\{E\left(\alpha_1, \alpha_2, \delta\right) : 0 < \alpha_1, \alpha_2, \delta \leq 1\}
$$

*is a set of events, such that*

*(i) For all $0 < \alpha \leq 1$ and $0 < \delta \leq 1$,*

$$
\Pr\left\{E\left(\alpha, \alpha, \delta\right)\right\} \leq \delta.
$$

*(ii) For all $0 < \alpha_1 \leq \alpha \leq \alpha_2 \leq 1$ and $0 < \delta_1 \leq \delta \leq 1$*

$$E\left(\alpha_1, \alpha_2, \delta_1\right) \subseteq E\left(\alpha, \alpha, \delta\right).$$

*Then for $0 < a, \delta < 1$,*

$$\Pr \bigcup_{\alpha \in (0,1]} E\left(\alpha a, \alpha, \delta \alpha \left(1 - a\right)\right) \leq \delta.$$

*Proof.* Define the events

$$E\left(\alpha_1, \alpha_2, \delta\right) := \left\{ \exists P, KL\left(P, Q_{\mathbf{X}}\right) \leq \alpha_2^{-1}, \mathbb{E}_{h \sim P}\left[\Delta\left(h, \mathbf{X}\right)\right] > \sqrt{\frac{\frac{3}{\sigma^2} \mathcal{V}\left(A\right) + \alpha_1^{-1} + \ln\left(1/\delta\right)}{n}} \right\}.$$

By Proposition B.6 they satisfy (i) of Lemma B.7 and it is easy to see, that (ii) also holds. If we set $a = 1/2$ the conclusion of Lemma B.7 becomes

$$\mathbb{E}_{h \sim P}\left[\Delta\left(h, \mathbf{X}\right)\right] > \sqrt{\frac{\frac{3}{\sigma^2} \mathcal{V}\left(A\right) + 2KL\left(P, Q_{\mathbf{X}}\right) + \ln\left(2KL\left(P, Q_{\mathbf{X}}\right)/\delta\right)}{n}}.$$

$\square$

To get a PAC-Bayesian bound analogous to Theorem 4.1 (iv) we proceed similarly and obtain the intermediate bound that for $\delta > 0$ with probability at least $1 - \delta$ in $\mathbf{X} \sim \mu^n$ for all $P$ such that $\mathbb{E}_{h \sim P}\left[v\left(h\right)\right] \leq V$ and $KL\left(P, Q_{\mathbf{X}}\right) \leq K$

$$\mathbb{E}_{h \sim P}\left[\Delta\left(h, \mathbf{X}\right)\right] \leq 2\sqrt{V\left(c^2 + \frac{\frac{3}{\sigma^2} \mathcal{V}\left(A\right) + K + \ln\left(1/\delta\right)}{n}\right)} + b\left(c^2 + \frac{\frac{3}{\sigma^2} \mathcal{V}\left(A\right) + K + \ln\left(1/\delta\right)}{n}\right).$$

Then we proceed as above, except that we have to use Lemma B.7 twice, once with $K$ and once with $nV$. The result of this mechanical procedure is

**Theorem B.8.** *For $\delta > 0$ with probability at least $1 - \delta$ in $\mathbf{X} \sim \mu^n$ for all $P$*

$$\mathbb{E}_{h \sim P}\left[\Delta\left(h, \mathbf{X}\right)\right] \leq 2\sqrt{\left(2\mathbb{E}_{h \sim P}\left[v\left(h\right)\right] + 1/n\right) \frac{\frac{3}{\sigma^2} \mathcal{V}\left(A\right) + C}{n}} + \frac{\frac{3}{\sigma^2} \mathcal{V}\left(A\right) + C}{n},$$

*where*

$$C = 2KL\left(P, Q_{\mathbf{X}}\right) + 1 + \ln\left(2\left(KL\left(P, Q_{\mathbf{X}}\right) + 1\right)\left(2\left(n\mathbb{E}_{h \sim P}\left[v\left(h\right)\right] + 1\right)\right)/\delta\right).$$

# C    Table of notation

| | |
|---|---|
| $\mathcal{X}$ | space of data |
| $\mu$ | probability of data |
| $n$ | sample size |
| $\mathbf{x}$ | generic member $(x_1, ..., x_n) \in \mathcal{X}^n$ |
| $\mathbf{X}$ | training set $\mathbf{X} = (X_1, ..., X_n) \sim \mu^n$ |
| $\mathcal{H}$ | loss class (loss fctn. composed with hypotheses, $h : \mathcal{X} \to [0, \infty))$ |
| $\mathcal{P}(\mathcal{H})$ | probability measures on $\mathcal{H}$ |
| $\pi$ | nonnegative a-priori measure on $\mathcal{H}$ |
| $L(h)$ | $L(h) = \mathbb{E}_{x \sim \mu}[h(x)]$, expected loss of $h \in \mathcal{H}$ |
| $\hat{L}(h, \mathbf{X})$ | $\hat{L}(h, \mathbf{X}) = (1/n) \sum_{i=1}^{n} h(X_i)$, empirical loss of $h \in \mathcal{H}$ |
| $\Delta(h, \mathbf{X})$ | $L(h) - \hat{L}(h, \mathbf{X})$, generalization gap |
| $Q$ | $Q : \mathbf{x} \in \mathcal{X}^n \mapsto Q_{\mathbf{x}} \in \mathcal{P}(\mathcal{H})$, stochastic algorithm |
| $Q_{\mathbf{x}}(h)$ | density w.r.t. $\pi$ of $Q_{\mathbf{x}}$ evaluated at $h \in \mathcal{H}$, $Q_{\mathbf{x}}(h) = \exp(H_Q(h, \mathbf{x}))$ |
| $H$ | $H : \mathcal{H} \times \mathcal{X}^n \to \mathbb{R}$, Hamiltonian |
| $Z$ | $Z : \mathcal{X}^n \to \mathbb{R}$, $Z(\mathbf{x}) = \int_{\mathcal{H}} \exp(H(h, \mathbf{x})) d\pi(h)$, partition function |
| $H_Q$ | $H_Q(h, \mathbf{x}) = H(h, \mathbf{x}) - \ln Z(\mathbf{x}) = \ln Q_{\mathbf{x}}(h)$ |
| $S_y^k$ | $S_y^k \mathbf{x} = (x_1, ..., x_{k-1}, y, x_{k+1}, ..., x_n)$, substitution operator |
| $D_{y,y'}^k$ | $(D_{y,y'}^k f)(\mathbf{x}) = f(S_y^k \mathbf{x}) - f(S_{y'}^k \mathbf{x})$, partial difference operator |
| $kl(p, q)$ | $kl(p, q) = p \ln \frac{p}{q} + (1-p) \ln \frac{1-p}{1-q}$, re. entropy of Bernoulli variables |
| $KL(\rho, \nu)$ | $\int \left(\ln \frac{d\rho}{d\nu}\right) d\rho$, KL-divergence of p.-measures $\rho$ and $\nu$ |
| $\|.\|$ | Euclidean norm on $\mathbb{R}^D$. |

