# OpenReview forum: "Generalization of Hamiltonian algorithms"
_NeurIPS.cc/2024/Conference — NeurIPS 2024 poster_

### Official Review · Reviewer_Dxrq · 2024-06-24

**Soundness:** 3
**Presentation:** 3
**Contribution:** 3
**Rating:** 6
**Confidence:** 4

**Summary:**

This work provides a novel approach to bound the generalization error (high probability bounds) of the Gibbs algorithm as an important example of Hamiltonian algorithms. The authors also applied their method to a different example of the Hamiltonian algorithm.

**Strengths:**

Strengths:

1- A novel approach is proposed for bounding the generalization error of the Gibbs algorithm.

2- There are some improvements in comparison with other works.

3- The proof approach is well-discussed.

4- Section about Randomization of stable algorithms is interesting.

**Weaknesses:**

1- In the case of the Gibbs algorithm, the contribution of this work is incremental. In particular, the authors should elaborate on why their results for the Gibbs algorithm are novel.

2- For the Gibbs algorithm, the main challenge is studying the asymptotic regime, where $\beta$ ---> \infty. It would be useful to study this regime or mention it as limitation of this analysis.

**Questions:**

Sea weaknesses.

**Limitations:**

Sea weaknesses.

---

> ### Author Rebuttal · Authors · 2024-08-03
>
> Please also see the general rebuttal for planned improvements to the paper.
>
> 1 - The improvements for the Gibbs algorithm go beyond constants. (4) is
> smaller than the inequality in the next display by a factor of $1/\sqrt{\ln
> \left( 1/\delta \right) }$. If the confidence parameter $\delta $ goes to
> zero, the quotient of the next display to (4) goes to infinity. Similarly
> (5) is better than the competing bounds by a factor of $1/\ln \sqrt{n}$. And
> even if logarithmic factors are considered irrelevant: does a simpler and more
> general method of proof, which in some special case gives only incrementally improved results, not provide
> a clearer perspective on the underlying phenomenon?
>
> 2 - It is planned to use an extra page for a future-work/limitations
> section. There will be reference to the challenge of the $\beta \rightarrow
> \infty $ limit, and the $\beta >n$ regime. These problems seem to require
> techniques different from those developed in the paper.

---

> > ### Comment · Reviewer_Dxrq · 2024-08-13
> > **Feedback**
> >
> > I want to thank the authors for their response.
> > I hope that the asymptotic discussion will be included in the final version.

---

### Official Review · Reviewer_hAd4 · 2024-07-13

**Soundness:** 2
**Presentation:** 2
**Contribution:** 2
**Rating:** 6
**Confidence:** 2

**Summary:**

Summary
-------

The papers presents a method to bound the `generalization gap', i.e the difference between
the expected and empirical losses for a hypothesis h drawn from a probability class
returned by the stochastic learning algorithm. The authors present a general-purpose
method to bound the exponentiated difference, and apply this technique to Gibbs sampling
(and other applications).

As someone who is not an expert on the topic, but still theoretically inclined, I was
unable to appreciate the contributions of this paper. It didn't help that the paper's
presentation was not up to scratch either (see comments below). I have given a
middle-of-the-line score to reflect this but will wait to hear from expert reviewers.



Detailed comments
=================

Questions
- Is h a hypothesis or a loss function? In 9, 10 and equation (1), it is written as a loss
  but in line 12 it is treated as a hypothesis?
- What is the motivation for bounding the difference between the expected loss of a
  a *learned* hypothesis, and the error on the training set? In typical use cases, one is
  interested in the test/generalization error after training.
- From what I could gather from the Gibbs sampling use case in 4.1, this can be used to
  bound the rate of convergence of Gibbs sampling. Is this correct?


The presentation of the paper could improve significantly.
- It is customary (at least in CS venues), to include an outline of the results, describe
  their significance of each result, and outline the key challenges and
  techniques used in the paper.
- The paper would also benefit from a dedicated related work section, where the authors
  explicitly state the most relevant work and how their results compare to them. The most
  interesting comparison I saw was in section 4.1 where they compare their bound to
  Rivasplata et al, and the improvement in the \sqrt{\log(1/\delta)} term. Even here, the
  authors have not explained why this gap \Delta is significant. Does it lead to fast
  convergence rates for Gibbs sampling?
  Perhaps the authors could state this result up front to prime the reader for what they
  can expect from this paper.
- On the same note, the authors should do a better job of highlihgting the key results in
  this paper. Currently, there are several lemmas/theorems presented one after the other
  and it is hard to appreciate which ones are the most significant, and why they are
  significant. My recommendation would be to identify the ~2 main theorems of the paper

**Strengths:**

See above

**Weaknesses:**

See above

**Questions:**

See above

**Limitations:**

See above

---

> ### Author Rebuttal · Authors · 2024-08-03
>
> Please also see the general rebuttal for planned improvements to the paper.
>
> Q1: $\mathcal{H}$ is a loss-class and $h\in \mathcal{H}$ is a hypothesis
> composed with a fixed loss function. Loss-classes are a notational
> simplification frequently used in the analysis of generalization.
>
> Q2: The bounds apply to the difference of expected and training error after
> training.
>
> Q3: The bounds apply to hypotheses sampled from the Gibbs distribution and
> give convergence in the sample size $n$. They do not refer to the
> convergence of Monte Carlo methods to the Gibbs distribution.

---

> > ### Comment · Reviewer_hAd4 · 2024-08-12
> > **Increasing score**
> >
> > Thanks for your response. Based on the rebuttal and the other reviews, I have increased my score.

---

### Official Review · Reviewer_16mk · 2024-07-14

**Soundness:** 4
**Presentation:** 4
**Contribution:** 3
**Rating:** 8
**Confidence:** 2

**Summary:**

This submission is a purely theoretical work, whose main goal is to bound $\Delta(h,\mathbf{X})=\mathbb{E}[h(X)]-\frac{1}{n}\sum^n_{i=1}h(X_i)$, i.e. equation (1). After some preliminary results, the results that show this bound under different conditions are Theorem 3.4, Theorem 3.5 and Theorem 3.8. Some possible (theoretical) applications are also shown, in the form of the Gibbs algorithm, randomisation of stable algorithms and PAC-Bayes bounds with data-dependent priors.

**Strengths:**

Unfortunately, I have not worked in any of the subfields that this paper is concerned with, so my judgment on its importance should be taken with a pinch of salt, but I am convinced of its significance and the contributions that this paper makes.

I have also tried to go through some of the maths in detail to check for its soundness, and barring a couple of very minor errors (see "Questions"), I think this paper is excellent in that regard.

I really liked the way this paper was written, straight to the point with minimum fuss, and my impression is overwhelmingly positive.

**Weaknesses:**

I think the authors made a deliberate choice to be concise with the proofs so that the proof can be included in the main body of the paper. I really like this, but in some places, the proof is too short, so that the readers are asked to do a significant amount of algebra by themselves. If possible, it would be great if the authors could use the extra page to flesh out the proofs a little bit.

**Questions:**

Displayed equation after L89: This seems to be a pointwise statement for both $h$ and $\mathbf{x}$, so should it be “for all $h\in\mathcal{H}$ and for all $\mathbf{x}\in\mathcal{X}^n$? Unless the statement should be “… is called a Hamiltonian for $Q_\mathbf{x}$” instead of “Hamiltonian for $Q$”?

Displayed equation after L162: Shouldn’t $2bc$ be $\frac{bc}{2}$, taking into account the factor of $\frac{1}{8}$? The $2nc^2$ in the numerator of the fraction should also be $\frac{nc^2}{2}$. It looks like this is corrected on L163, and working through the maths, I indeed got the claimed inequality in Theorem 3.4(ii).

L183: I think you are missing a $\ln$ in front of the exponential?

**Limitations:**

L88: "A function $H:\mathcal{H}\times\mathcal{X}^n$" perhaps it is better to write “A function $H$ on $\mathcal{H}\times\mathcal{X}^n$? I leave this up to the authors.

Displayed equation after L153: The spacing is a bit strange on the left?

L185: an -> and

Displayed equation after L195: The comma should be a full stop.

L178, L389: “Assume that all” -> “Assume that for all”

L391: everz -> every

---

> ### Author Rebuttal · Authors · 2024-08-03
>
> Special thanks to you for the encouraging words and the careful reading,
> which uncovered several typos and inaccuracies.
>
> Please also see the general rebuttal for planned improvements to the paper.
>
> Question 1: L89. Yes, it should be "for all $h\in \mathcal{H}$ and $\mathbf{x%
> }\in \mathcal{X}^{n}$".
>
> Question 2: Display after L162. You are absolutely right, thank you. No idea
> how this got into the paper.
>
> Question 3: L183. You are right again, thank you.
>
> L88: Will be changed. The other typos will be corrected accordingly.

---

> > ### Comment · Reviewer_16mk · 2024-08-14
> >
> > Dear authors,
> >
> > Thank you for your response. I have read them, also the other reviews, and I retain my positive evaluation of this submission.

---

### Official Review · Reviewer_mcjd · 2024-07-18

**Soundness:** 4
**Presentation:** 4
**Contribution:** 3
**Rating:** 6
**Confidence:** 4

**Summary:**

This paper introduces a general method to bound the logarithm of the expecattion of the exponential of the generalization gap for stochastic learning algorithms. This method is applicable when the distribution of the algorithm concentrates exponentially around its mean, extending to cases where the Hamiltonian form satisfies a bounded difference condition or is sub-Gaussian. Further, the paper discuss applications to the gibbs algorithm, improving existing bounds, Lastly the paper extends generalization guarantees to hypotheses sampled once from stochastic kernels centered at the output of uniformly stable algorithms. This advancement strengthens previous results and improves the understanding of generalization properties for algorithms based on stochastic kernels.

**Strengths:**

The paper is very-well written. it is crisp and clear about the notations, contributions and the context of the novel results with respect to the exsiting literature. The theoretical rigor is well-presented and the results are very well supported through coherent arguments. I specially appreciate the "self-contained" nature of the paper, where everything including the very fundamental markov inequality is written in the appendix for convenience of the reader so there is no need to refer to other papers to understand, verify and appreciate the contributions of the paper.

The paper makes non-trivial theoretical contributions. It provides generalization guarantees for bounded hypothesis (section 3.1) and unbounded hypothesis through subgaussianity (Section 3.2). The contributions include a general method for bounding generalization gaps for the specific case of Hamiltonian algorithms, thereby identifying an important subclass of problems to bound the notorious ln E_X E_h [exp \lambda \Delta] which is often not amenable for general problems. This contribution is important. Further, the authors specify precisely where the bounds improve upon exisiting results (e.g. dependence on the confidence parameter \delta is improved from previously known O(\sqrt(1/\delta) to O(1/\delta) for the gibbs algorithm), often with simpler proof techniques and cleaner constants/exponents.

**Weaknesses:**

The paper is very notation-heavy, and likely not accessible to everyone. This is not really a critique of the paper tbh, but more of the nature of the result/paper and the short conference format. The authors have done a fair job of explaining the notations. But It is easy to get lost in keeping track. For example, the notion of canonical Hamiltonian H_Q and its usefulness (for example in the simple proof of Prop 3.1) is easy to overlook unless one really gets into the weeds. I would suggest adding a couple of sentences cementing the importance of H_Q in the analysis. Beyond helping through and simplifying some results, is there additional importance of H_Q ?

The conclusion and future directions/discussions/limitations section is severely lacking. Can the authors talk about limitations of the proposed method, and any further possible open problems that the reader might be interested in.

There is no empirical verification, even with toy setups. Can the authors comment on how the presented theory can be empirically demonstrated/verified?

**Questions:**

In addition to the weaknesses:

In context of the paper, and the definition of \Delta(h,X), which we know is centered, I am not sure how Prop 3.1 is useful or applied in this paper. I understand it could be potentially useful where F and H_Q can be handled separately (assuming F being non-centered is still somehow amenable to this). But for this paper, this is not the case.

The notation S^k_X f(\mathbf{x}) used in prop 3.2 onwards is clunky at best, and badly overloaded at worst. May I suggest using f(S^k_X \mathbf{x}) ?

theorem 3.4 needs h(y’) \in [0,b] ? I think you mean \forall t \in \mathcal{X} h(t) \in [0,b] ?

“In practice Q is often defined by specifying some Hamiltonian H , so HQ (h, x) = H (h, x) − ln Z (x) in general” – I am not sure what this sentence means. Please expand. Are you saying the canonical H_Q can be trivially written like that? How is Q being defined through that?
L186 “generalization guarantee of rapid convergence” .. could you expand on this please? What order is considered as “rapid”?

Minor: In a couple of places some of the references seem to be hard-coded ? e.g. references to A.1 because those are not clickable links.

**Limitations:**

Partially.

---

> ### Author Rebuttal · Authors · 2024-08-03
>
> Please also see the general rebuttal for planned improvements to the paper.
>
> Experiments to demonstrate the theoretical results are planned for future
> work. For real data the costly part is the repeated
> sampling from the Gibbs distribution, because one has to
> await the mixing time between Monte-Carlo-samples for approximate independence. Recording for each sample
> the training error and a test error for independent data provides the
> information to test the predictions on both the expected and un-expected
> generalization gaps, and their dependence on the training error. Note that
> the benefit of the un-expected bounds, once one is willing to accept their
> correctness, is that only one sample is needed in the test.
>
> Question 1: A separate treatment of $F$ and $H_{Q}$ is used in the proofs of
> the Bernstein-type inequality Theorem 3.5 and Theorem 3.8 (ii). Proposition
> 3.1 is essential in both of these cases.
>
> Question 2: The notation $S_{X}^{k}f\left( \mathbf{x}\right) $ has entered
> the paper through some copy/paste process. It is essentially a typo for
> which we apologize. As you say, $f\left( S_{X}^{k}\mathbf{x}\right) $ should
> be used.
>
> Question 3: In the statement of Theorem 3.4 the universal quantifier was
> understood to apply to \textit{all} of $k,h,y,y^{\prime }$ and $\mathbf{x}$.
> While this still seems correct, it is probably clearer to state the uniform
> boundedness of the $h\in \mathcal{H}$ separately, as you propose.
>
> Question 4: "In practice...". An example is the Gibbs-algorithm. One
> specifies $H\left( h,\mathbf{x}\right) =-\left( \beta /n\right)
> \sum_{i}h\left( x_{i}\right) $. Then the density is $Q_{\mathbf{x}}\left(
> h\right) =Z\left( \mathbf{x}\right) ^{-1}\exp \left( H\left( h,\mathbf{x}%
> \right) \right) =\exp \left( H\left( h,\mathbf{x}\right) -\ln Z\left(
> \mathbf{x}\right) \right) =\exp \left( H_{Q}\left( \mathbf{x}\right) \right)
> $.
>
> Question 5: "rapid convergence". What was meant is that then $\Delta \left( h,%
> \mathbf{X}\right) $ is approximately of order $1/n$, rather than $\sqrt{1/n}$
> as in Theorem 3.4. But, as this also depends on the order which $c$ has in $%
> n $, it really depends on the concrete application of Corollary 3.7. Since
> "convergence" may also be confused with the convergence of a
> Monte-Carlo-method to its limiting distribution, the text will be modified.

---

> > ### Comment · Reviewer_mcjd · 2024-08-09
> > **Thank you for your response**
> >
> > The authors have clarified my minor doubts and I confirm my opinion that this is a solid theoretical contribution. Some experiments, even simulated ones, to validate the results will improve the quality of the paper even further.

---

### Author Rebuttal · Authors · 2024-08-03

Many thanks to the reviewers, who provided many useful comments. Major
planned improvements are:

1. The revision will contain a section on future directions and
limitations. It will mention potential applications to iterated algorithms
and weakly dependent data. The limitations part will point to the remaining
challenges of the $\beta >n$ regime and the limit $\beta \rightarrow \infty $.
.

2. A glossary of notation in tabular form will be included at the
beginning of the appendix.

---

### Decision · Program_Chairs · 2024-09-25

**Decision:**

Accept (poster)

**Comment:**

This paper introduces a method for bounding the generalization gap of stochastic learning algorithms by focusing on the exponential concentration of the algorithm's distribution around its mean. The approach is applicable to various scenarios, including Gibbs sampling and randomizations of stable algorithms, enhancing the understanding and existing bounds on generalization properties. The reviewers unanimously praised the paper for its clarity, theoretical rigor, and mathematical correctness, with concerns adequately addressed in the rebuttal. Given its strong theoretical contributions, I recommend acceptance, in agreement with all four reviewers.